# Pulsed electrosynthesis orthogonally optimizes C–N coupling and hydrogenation for amine production with a molecular catalyst

Shuai Yan[1], Yang Wang[1], Shuai Chen [1], Jost Heise [1], Morgan McKee [1], Chen Gao[1], Xiaodong Li [2,3] ✉ & Nikolay Kornienko [1] ✉

Co-electrolysis of $CO_2$ and nitrate offers a sustainable route to organic amines but suffers from a kinetic mismatch between C-N coupling and hydrogenation steps under static conditions. This mismatch is challenging to address through conventional catalyst design and therefore limits both efficiency and selectivity. Here, we introduce a pulsed strategy that orthogonally decouples these steps by alternating optimized potentials. Pulses at less reductive potentials suppress hydrogenation and thus favor oxime formation, whereas more reductive potentials promote hydrogenation to amines. Using cobalt phthalocyanines, this approach triples the reaction rate and doubles the selectivity for methylamine compared to static methods, and also enables the formation of higher amines. In situ studies and density functional theory calculations reveal that a more reductive pulse accelerates hydrogenation, promoting a multielectron cascade through intermediates. Retrosynthetic analysis and product distribution trends further support a sequential coupling-hydrogenation pathway from methylhydroxylamine/methylamine to higher amines. This work offers a framework for steering multistep C–N bond formation and shows how dynamic electrochemistry can turn waste-derived carbon and nitrogen into valuable products.

Anthropogenic $CO_2$ emissions reached 41.6 gigatons in 2024, representing a 2% increase compared to 2023 levels and further contributing to the greenhouse effect[1]. Meanwhile, high levels of $NO_3^-$ are a major contributor to water pollution and eutrophication[2]. Instead of treating these species only as pollutants, their electrochemical co-reduction offers a sustainable route towards carbon- and nitrogen-neutral manufacturing by upcycling waste into valuable chemicals such as urea, amines, oximes, and amides, which are essential to agriculture, medicine, and the chemical industry[3–7]. Among these products, amines and oximes are highly desirable but remain

challenging targets for electrochemical synthesis, with only limited reports demonstrating their formation from $CO_2$ and $NO_3^{-[4,8–11]}$.

A primary challenge is the kinetic mismatch inherent in the reaction mechanism (Fig. 1a). The process typically proceeds through the reduction of $CO_2$ and $NO_3^-$ to reactive intermediates, followed by C–N bond formation and subsequent hydrogenation to produce amine products[12]. Although the pre-activation of the reactants and subsequent coupling step is usually fast, hydrogenation involves multiple sluggish electron/proton transfers and is considerably more challenging[13,14]. The challenge is further amplified for multi-carbon or

[1]Institute of Inorganic Chemistry, University of Bonn, Bonn, Germany. [2]Max Planck Institute of Microstructure Physics, Halle, Germany. [3]Key Laboratory of Precision and Intelligent Chemistry, University of Science and Technology of China, Hefei, China. ✉e-mail: Xiaodong.li@tu-dresden.de; nkornien@uni-bonn.de

**Fig. 1 | General concept. a** Electrochemical C–N coupling mechanism from $CO_2$ and $NO_3^-$. **b** Pulsed techniques for electrosynthesis. **c** More or less reductive pulse orthogonally decouples the coupling and hydrogenation step to enable oxime and amine formation, respectively.

multi-amine products—for example, trimethylamine formation is estimated to require over 27 hydrogenation steps. Previous strategies to promote hydrogenation, such as introducing proton-shuttling ligands[15], hydrogenation-active metals (e.g., Pd, Ni, Pt)[16–18], or operating in acidic electrolytes[19] often induce undesired side reactions and even compromise catalyst stability. Moreover, static electrolysis offers limited temporal control over coupling and hydrogenation[20]. Dynamic potential control emerges as a promising alternative by temporally decoupling these steps, especially as hydrogenation steps strongly depend on applied potential[21,22]. Redox pulse[23,24] or less reductive pulse[25–27] have been shown to enhance coupling kinetics through periodic modulation of the applied potential (Fig. 1b). In particular, applying deeper cathodic potentials can increase surface hydrogen coverage and drive hydrogenation more efficiently, as seen in the conversion of oximes to amines, which requires more negative potentials than the initial coupling step[28]. Building on these insights, we thus propose that alternating potentials in pulsed electrolysis allow selective control over hydrogenation, enabling access to either partially or fully reduced products depending on the applied pulse potential (Fig. 1b).

In this work, we develop a more reductive pulsed electrolysis that regulates the hydrogenation step to enhance selectivity toward amines (Fig. 1c). At an optimal negative potential, C–N precursors selectively accumulate on the catalyst surface. A subsequent pulse to a more negative potential provides the additional thermodynamic driving force required to accelerate both the coupling of surface intermediates and their hydrogenation, thereby enhancing the overall flux through the C–N coupling pathway and increasing amine formation. In contrast, continuous electrolysis at a deeply negative potential accelerates both desired and competing reduction pathways simultaneously, preventing the selective buildup of reactive intermediates and resulting in lower overall selectivity. Using cobalt phthalocyanine molecules anchored on carbon nanotubes (CoPc/CNTs) as a model

catalyst, we show that pulses at less reductive potentials suppress hydrogenation and favor oxime formation. On the other hand, pulses at more negative potentials promote deeper hydrogenation to amines. Under more reductive pulses, a Faradaic efficiency (FE) of 13.7% and a partial current density of 9.8 mA cm$^{-2}$ are achieved for methylamine, with the formation of higher amines. Mechanistic studies demonstrate that deep reduction potentials are essential to accelerate the hydrogenation of key coupling intermediates. Retrosynthetic analysis indicates a sequential pathway, where methylhydroxylamine/methylamine form and then convert to higher amines (Fig. 1c). These findings establish dynamic potential control as a general strategy for steering product distributions across multiple hydrogenation steps in electrosynthesis.

## Results and discussion
We chose CoPc/CNTs as the catalyst as it has a well-defined structure suitable for mechanistic studies, can efficiently catalyze $CO_2$-to-HCHO and $NO_3^-$-to-$NH_2OH$ conversion[29,30], and enables subsequent methylamine formation[9]. CoPc alone is poorly conductive, and simple drop-casting can lead to aggregation, limiting electron transfer and reactant access[31]. Loading CoPc onto multiwalled carbon nanotubes (CNTs) ensures molecular-level dispersion, efficient electrical connection, and high reactant accessibility[31,32]. Molecularly dispersed CoPc was then synthesized on CNTs through strong π–π interactions, enabling uniform anchoring at the molecular scale[18]. Transmission electron microscopy (TEM, Supplementary Fig. 1) revealed that the CoPc/CNT catalyst retained the intrinsic tubular morphology of CNTs, with an average diameter of ~10 nm, while X-ray diffraction (XRD) confirmed the absence of CoPc aggregation (Supplementary Fig. 2). Scanning electron microscopy combined with energy-dispersive X-ray spectroscopy (SEM-EDX) mapping demonstrated a homogeneous Co and N distribution (Supplementary Fig. 3). A redshift in the Q-band of the UV–Vis spectrum indicated electronic interaction between CoPc and

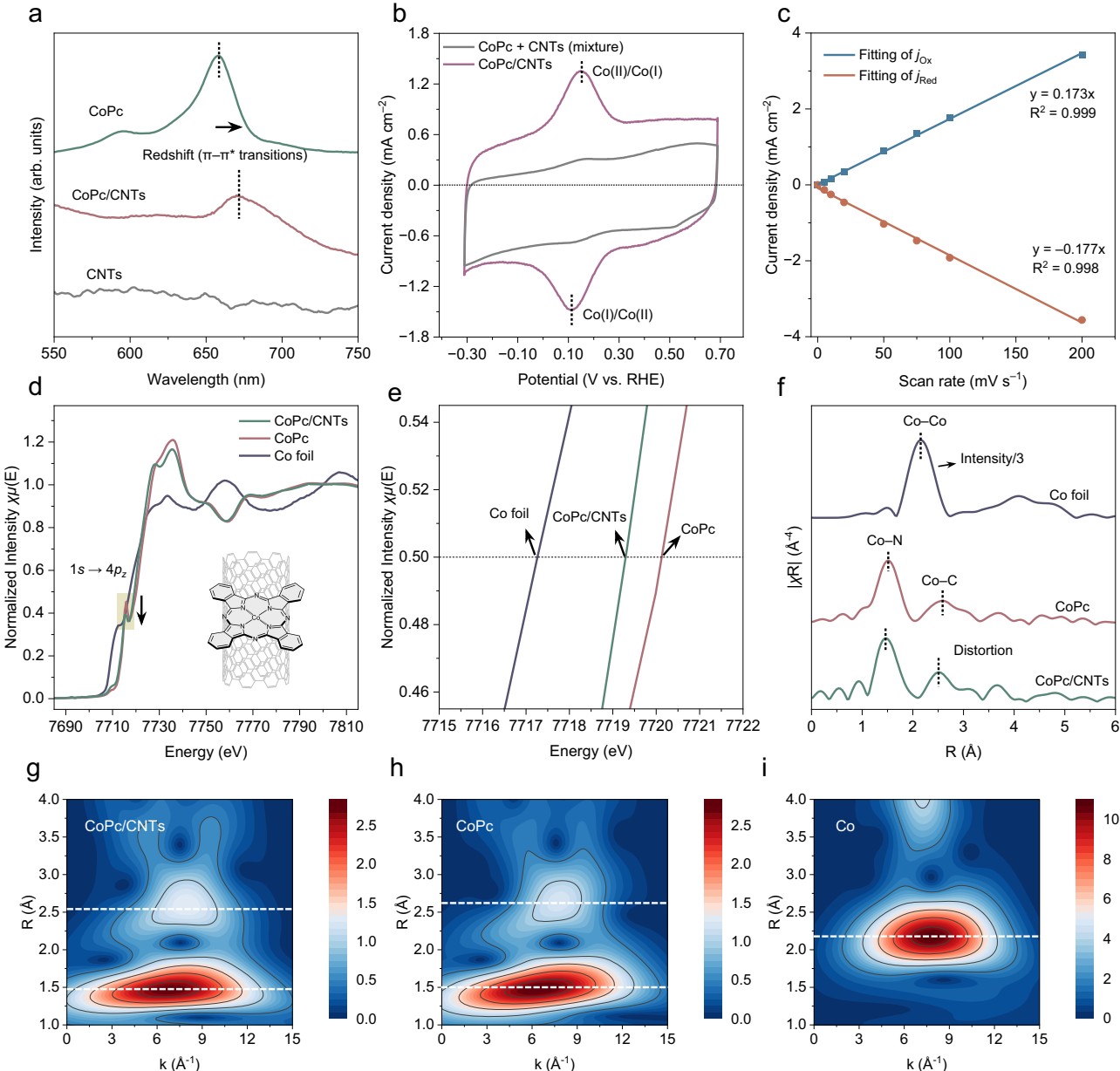

**Fig. 2 | Structural characterization. a** UV–Vis absorption spectra of CoPc and CoPc/CNTs in DMF. **b** Cyclic voltammetry (CV) curves of CoPc and CoPc/CNTs measured at 75 mV s⁻¹ in Ar-saturated 0.5 M KHCO₃. **c** Linear relationship between peak current density and scan rate for CoPc/CNTs derived from CV data. **d, e** Co K-edge XANES spectra and **f** Fourier-transformed R-space of the Co K-edge EXAFS spectra. Wavelet transform analysis for the $k^3$-weighted Co K-edge EXAFS of **g** CoPc/CNTs, **h** CoPc, and **i** Co foil. Source data are provided with this paper.

CNTs, indicating electron donation from CNTs that alters the electronic structure by narrowing the HOMO–LUMO gap (Fig. 2a)[33]. Electrochemically, CoPc/CNTs exhibited a sharp and reversible Co(II)/Co(I) redox couple with a narrow peak separation and linear scan rate dependence, indicating surface-confined, fast electron transfer (Fig. 2b, c and Supplementary Fig. 4)[33–35]. Compared to the physical mixture of CoPc and CNTs, CoPc/CNTs show stronger redox peaks (Fig. 2b), suggesting a larger fraction of electrochemically active cobalt enabled by molecular dispersion of CoPc on the CNTs[33].

X-ray absorption spectroscopy (XAS) measurements were utilized to investigate the electronic structure and coordination environment of CoPc/CNTs. The X-ray absorption near edge structure (XANES) spectra show CoPc/CNTs retain a $D_{4h}$-like symmetry comparable to pristine CoPc (Fig. 2d)[36]. However, dispersion on CNTs leads to a marked suppression of the pre-edge peak at -7715.8 eV, attributed to the $1s \rightarrow 4p_z$

transition characteristic of square-planar Co coordination[37,38]. This attenuation suggests electron back-donation from CNTs to Co $3d_{z^2}/4p_z$ orbitals, resulting in a symmetry change from $D_{4h}$ to $C_{4v}$[38]. Additionally, a 0.83 eV shift in the half-maximum edge position confirms electron transfer from CNTs to CoPc (Fig. 2e), consistent with UV–Vis observations (Fig. 2a). Extended X-ray absorption fine structure (EXAFS) analysis provides atomic-level insights. The Fourier-transformed $k^3$-weighted spectra (Fig. 2f) and wavelet transform (WT) analysis revealed a clear Co–N coordination peak, with no detectable Co–Co signals, confirming atomic dispersion of Co centers (Fig. 2g–i). Notably, Co–N and Co–C coordination distances contracted from 1.52 to 1.46 Å and 2.59 Å to 2.50 Å (Fig. 2f), respectively, indicating deformation of the phthalocyanine macrocycle due to interaction with CNTs.

Electrochemical C–N coupling was investigated in our home-made H-type cells using a CoPc/CNTs catalyst (Supplementary Fig. 5). All

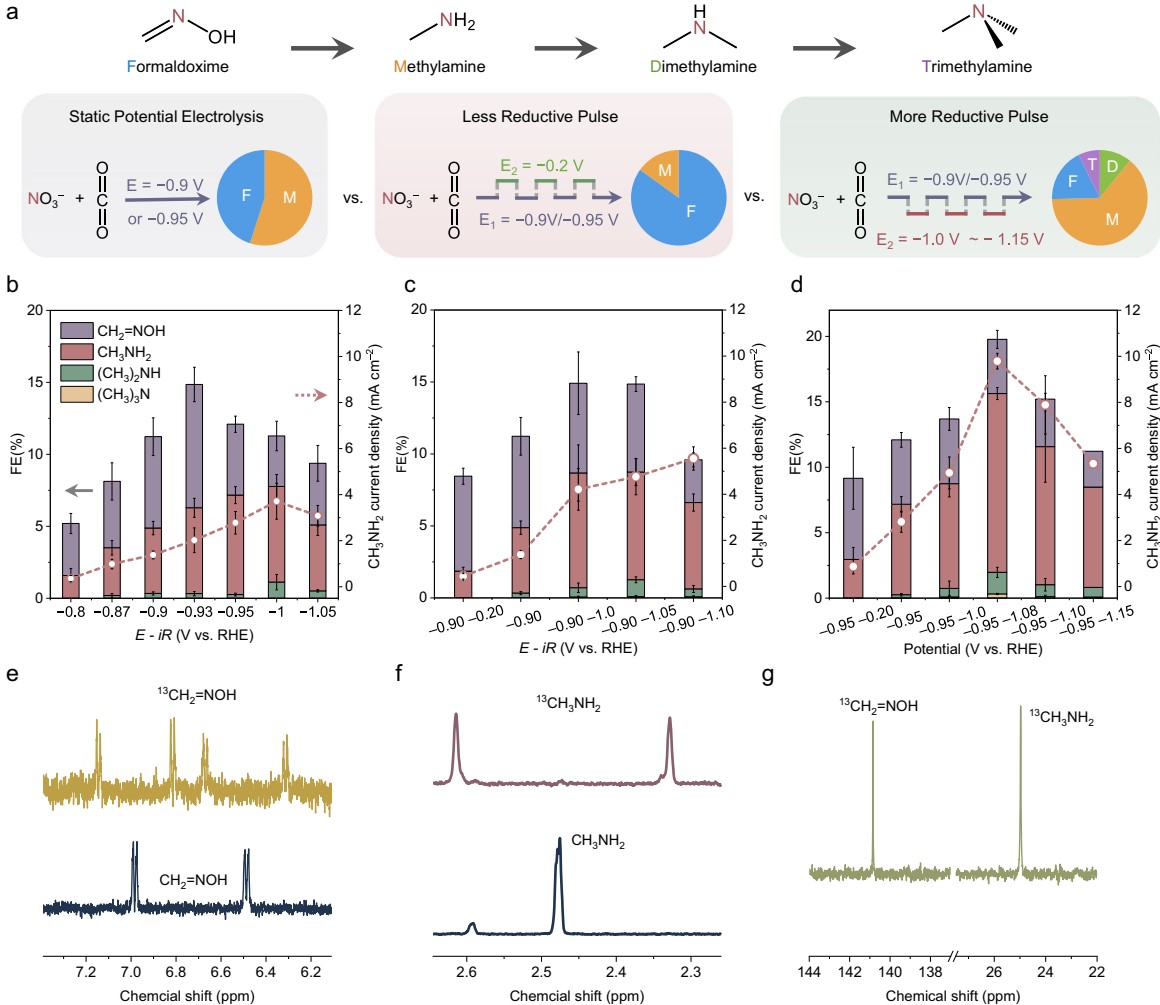

**Fig. 3 | Static electrolysis vs. pulsed electrolysis. a** Schematic illustration of selective C–N product formation under different electrolysis modes. The pie charts indicate general trends in product distribution (F, M, D, and T) as a function of the reductive pulse potential. (F: formaldoxime; M: methylamine; D: dimethylamine; T: trimethylamine). The Faradaic efficiencies (FEs) for C–N products and partial current density of methylamine under **b** static potentials and **c, d** pulsed potentials.

**e, f** $^1$H-NMR and **g** $^{13}$C-NMR spectra of C–N products. Reported potentials are corrected for 85% of the ohmic resistance (iR-correction). Electrolyte resistance and pH of $CO_2$-saturated 0.1 M $KHCO_3$ and 0.5 M $KNO_3$ electrolyte were measured as $16.0 \pm 0.5 \, \Omega$ and $6.8 \pm 0.1$, respectively, with an electrode surface area of 0.25 cm$^2$. Data are presented as mean $\pm$ s.d. ($n = 3$ independent experiments); error bars represent s.d. Source data are provided with this paper.

applied potentials were referenced to the reversible hydrogen electrode (RHE) with 85% $iR$ compensation. As shown in the CV curves (Supplementary Fig. 6), introducing $CO_2$ suppressed both $NO_3^-$ reduction and hydrogen evolution, lowering the overall current density and thereby favoring C–N bond formation. We next explored how pulsed electrolysis influences product distribution compared to static electrolysis (Fig. 3a). Static electrolysis at $E_1 = 0.90/0.95$ V ($t_1 = 2$ s) served as the reference point while $E_2$ was systematically varied with a pulse duration of 3 s (Supplementary Fig. 7). Gas and liquid products were quantified via gas chromatography (GC) and $^1$H NMR spectroscopy (Supplementary Fig. 8-11). A calibration curve for methylamine (0–1 mM) was prepared as a representative example (Supplementary Fig. 12), giving concentrations comparable to those obtained using the internal standard (IS) approach. The IS approach was therefore used throughout this work, as many amine and oxime products are unstable and isotopically labeled standards are unavailable.

Under static electrolysis, methylamine and formaldoxime were the major C–N products, with methylamine FE typically below 7% and partial current densities under 4.5 mA cm$^{-2}$ (Fig. 3b and Supplementary Fig. 13). By modulating the reductive potential, pulsed electrolysis enabled a tunable shift in selectivity, as illustrated by the trend in the

pie charts of Fig. 3a. Less reductive pulse ($E_2 = -0.2$ V) enhanced oxime formation and suppressed over-hydrogenation, increasing the oxime-to-amine selectivity ratio to ~3 times the original value (Supplementary Fig. 14). In contrast, more reductive pulse promoted deeper hydrogenation toward amines. For example, under $E_1 = -0.90$ V and $E_2 = -1.0$ V, the FE and partial current density for methylamine increased by 1.8- and 3.1-fold, respectively, compared to static operation (Fig. 3c and Supplementary Fig. 13). At an $E_2$ of $-1.08$ V, methylamine FE reached 13.7% with a current density of 9.8 mA cm$^{-2}$ (Fig. 3d). In particular, under these optimized pulsing conditions, higher-order amines formed, with dimethylamine and trimethylamine achieving FEs of 1.7% and 0.3%, respectively.

Compared with static electrolysis, C–N product selectivity is higher under more reductive pulses (Supplementary Fig. 15). The improvement in C–N coupling is primarily achieved by suppressing the side hydrogen evolution reaction (HER) at less negative potentials, which allows key intermediates to accumulate on the catalyst surface. During the more reductive periods, the higher overpotential accelerates the coupling of surface intermediates and their hydrogenation toward C–N products. In contrast, during the less reductive intervals, surface hydrogen coverage is partially relieved, thereby avoiding

continuous conditions that favor side products such as $H_2$ and $NH_3$. For example, at a static potential of −1.05 V, the FEs of $H_2$ and $NH_3$ are 23.3% and 12.3%, respectively, whereas pulsed operation between −0.95 and −1.08 V lowers them to 16% and 7.2% (Supplementary Fig. 16). Thus, while the overall current increases, C−N selectivity is also enhanced, leading to higher partial current densities of the desired products. However, applying more negative $E_2$ potentials led to a decline in both selectivity and activity for all C−N products, resulting in increased formation of side products due to excessive hydrogen coverage (Supplementary Fig. 10).

We also investigated the effect of pulse duration on product distribution and found that both longer (1 s/5 s) and shorter (5 s/1 s) pulses gave lower performance than the optimized 2 s/3 s sequence (Supplementary Fig. 17). Longer pulses led to decreased oxime selectivity, suggesting a balance between intermediate formation and side reactions. These findings demonstrate that pulsed electrolysis provides a powerful handle to control hydrogenation depth, enabling selective access to oximes, monoamines, and higher-order amines. Isotopic labeling experiments using $^{13}CO_2$ and $NO_3^-$ confirmed that the liquid C−N products were from the intended reactants, as evidenced by post-electrolysis $^1H$-NMR and $^{13}C$-NMR (Fig. 3e–g). The $^1H$ signals of the products exhibit characteristic doublets due to $^1H$–$^{13}C$ coupling $(^1J_{CH})$[39,40], confirming that the carbon atoms in the products originate from $^{13}CO_2$[9,39,40].

Moreover, under the tested conditions, pulsed electrolysis maintained a relatively stable performance for methylamine production over multiple cycles (Supplementary Fig. 18), providing a valuable model for exploring how pulse conditions influence hydrogenation and C−N coupling. When the electrolysis time was extended to 3 h, the selectivity for methylamine decreased; however, all performance comparisons and mechanistic analyses in this work were conducted within the stable operation window of 0.5 h. Post-reaction characterization confirmed that the CoPc/CNT catalyst largely retained its structural integrity and atomic dispersion, with no evidence of aggregation or cobalt leaching (Supplementary Figs. 19–21 and Supplementary Table 1). The observed decrease in methylamine selectivity over extended operation may be due to partial catalyst detachment or subtle surface changes not captured by bulk characterization.

To further investigate whether pulsed electrolysis also promotes higher amine formation on other catalysts, we evaluated a Cu catalyst in a home-made gas diffusion electrode cell. Under static potentials of −0.4 V and −1.0 V, only trace methylamine was detected and no dimethylamine was observed (Supplementary Fig. 22). In contrast, applying pulsed potentials (−0.4 V for 10 s and −1.0 V for 2 s) increased methylamine production and enabled the formation of dimethylamine. These observations demonstrate that pulsed electrolysis enhances the generation of key C−N coupling intermediates and accelerates hydrogenation steps, consistent with the mechanistic insights obtained from CoPc/CNTs. We further extended the reaction to $CH_3NO_2$ and $CO_2$ substrates. The use of optimized reductive pulses increased the partial current density of dimethylamine (Supplementary Fig. 23).

To investigate the effect of potentials on hydrogenation efficiency, we used formaldoxime as a substrate. The addition of formaldoxime increased the total current density, indicating enhanced electrochemical activity (Fig. 4a). As the overpotential increased, the FE for N-methylhydroxylamine gradually declined, while that for methylamine increased and then decreased (Supplementary Fig. 24). Similarly, Fig. 4b shows that the partial current density for N-methylhydroxylamine remained relatively constant, whereas that for methylamine increased continuously, reaching its peak at –1.0 V. These observations suggest that more negative potentials facilitate the initial hydrogenation of formaldoxime to N-methylhydroxylamine and its further reduction to methylamine. Kinetic isotope effect (KIE) studies demonstrate that proton transfer is involved in the rate-determining steps. The suppression of amine formation in the deuterated electrolyte confirms that high overpotentials enhance intermediate protonation, driving full hydrogenation to methylamine (Fig. 4c, d). At excessively negative potentials, however, high hydrogen coverage can suppress C−N bond formation, reducing product selectivity, as further reflected by the N-methylhydroxylamine/methylamine current density ratio (Fig. 4b).

Building on the performance observed under pulsed electrolysis, we turned our attention to understanding the mechanistic origins of amine selectivity. In particular, we aimed to clarify how dynamic potential modulation impacts the formation and transformation of key intermediates involved in C−N coupling. To this end, we utilized in-situ attenuated total reflection surface-enhanced infrared absorption spectroscopy (ATR-SEIRAS) to detect potential reaction intermediates (Supplementary Fig. 25). Using the system at open circuit conditions as a background, we observed bands at 1285 cm$^{-1}$ under C−N coupling conditions, which match N–O stretching of $^*NO_2$ (Fig. 5a and Supplementary Table 2)[41]. As the applied potential became more negative (beyond −0.8 V), additional bands appeared at around 1412 and 1690 cm$^{-1}$, which could be attributed to the C−N stretching and C=N stretching vibrations, respectively[42,43]. Interestingly, the intensity of the C=N band peaked at −1.0 V, but disappeared centered at −1.2 V. In contrast, the C−N band increased with increasing overpotential, suggesting that deeper reduction facilitates the hydrogenation of C=N intermediates into C−N products. Control experiments on $NO_3^-$ reduction (Supplementary Fig. 26) show that the observed C−N signals originate from C−N formation rather than $NO_3^-$. To further validate our peak assignments, we performed isotope $^{15}N$-$NO_3^-$ labeling experiments, which showed a redshift in the corresponding vibrational bands (Fig. 5b), consistent with the expected behavior upon substitution with heavier atoms[40]. In the $^{15}N$-labeled spectra, $^{15}N$ substitution is expected to lower the C=N stretching frequency, likely shifting it into the strong absorption region of $H_2O$ bending vibrations and thus obscuring the signal[44,45]. To reduce this interference, $D_2O$ was used as the solvent, allowing the C=$^{15}N$ signal to be observed clearly and confirming its assignment (Supplementary Fig. 27). We also investigated the potential influence of pulsed electrolysis on the hydrogen-bond network of interfacial water using in-situ ATR-SEIRAS (Supplementary Fig. 28). Only minor differences are observed compared with static electrolysis, indicating that the effects of pulsed operation on hydrogenation and amine formation are more likely related to surface reaction dynamics than to changes in the interfacial water structure.

To provide further insights into the influence of potentials on C−N coupling, in-situ electrochemical mass spectrometry (ECMS) was used to monitor volatile intermediates and products with $CO_2$ and $NO_3^-$ as reactants (Supplementary Fig. 29). Because $CO_2$ is continuously consumed during the reaction rather than suddenly generated electrochemically, its signal does not appear as a sharp peak; instead, we monitored its concentration in real time throughout the process (Supplementary Fig. 30). At −0.53 V, the $CO_2$ signal exhibited a sharper decline[46], accompanied by the appearance of CO and CHO species (Fig. 5c and Supplementary Table 3). CO reached its maximum intensity at −0.75 V, after which its signal decreased due to continuous $CO_2$ consumption and possible conversion to downstream intermediates. Interestingly, the CHO signal started to increase at more negative potentials, following the formation of CO. The CHO intensity plateaued around −0.86 V and then increased gradually at more negative potentials, suggesting that higher overpotentials facilitate the hydrogenation of CO to CHO (Fig. 5c). A similar formation potential window was observed for $NH_2OH$, consistent with the coupling of formaldehyde and hydroxylamine to produce formaldoxime. As the potential became even more negative, the intensities of $H_2$ and $NH_3$ continued to increase, reaching a maximum near −1.0 V (Fig. 5c). This

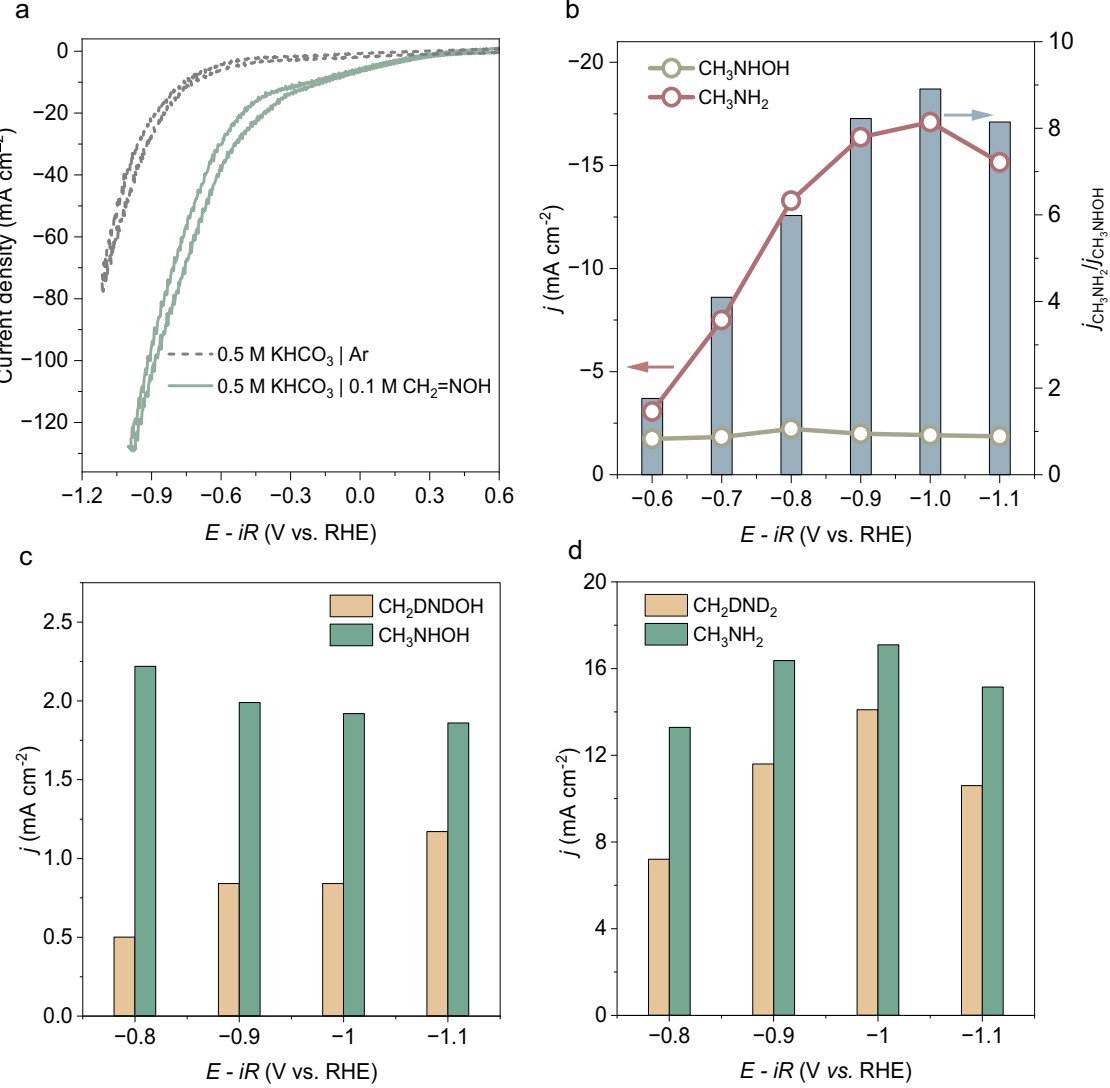

**Fig. 4 | Formaldoxime hydrogenation performance. a** CVs of the CoPc/CNTs showing formaldoxime reduction. **b** Partial current densities ($j$) of CH₃NHOH and CH₃NH₂, and the corresponding ratio of $j_{CH3NH2}/j_{CH3NHOH}$ at various potentials. Kinetic isotope effect (KIE) for the formation of **c** CH₃NHOH/CH₂DNDOH and

**d** CH₃NH₂/CH₂DND₂. Reported potentials are corrected for 85% of the ohmic resistance (iR-correction). Electrolyte resistance and pH were measured as $16.4 \pm 0.3\,\Omega$ and $8.3 \pm 0.2$, respectively, with an electrode surface area of $0.25\,cm^2$. Source data are provided with this paper.

suggests that overly reductive potentials may promote competing reactions and suppress C–N product formation.

To elucidate the reaction mechanism underlying the formation of amines, we conducted retrosynthetic analysis using key C–N coupling products as reactants. Electrolysis in CO₂-saturated KHCO₃ and methylamine electrolytes produced dimethylamine predominantly, with minor trimethylamine (Fig. 5d). Using dimethylamine instead led to trimethylamine as the main product (Fig. 5d and Supplementary Fig. 31). To investigate the second C–N bond formation, we carried out C–N coupling experiments using methylhydroxylamine, formaldoxime, and methylamine with formaldehyde (Supplementary Fig. 32). We found that using methylhydroxylamine as the precursor showed the highest dimethylamine formation rate, followed by methylamine and formaldoxime. These results indicate that the second C–N bond predominantly originates from methylhydroxylamine, with methylamine also contributing, as reflected by its high abundance in the products.

Building on the mechanistic insights obtained from electrocatalytic and in situ analyses, we performed density functional theory (DFT) calculations to investigate how applied potentials modulate

hydrogenation and C–N bond formation. In our calculations, CoPc supported on 10 nm-diameter carbon nanotubes (from TEM images, Supplementary Fig. 1) was constructed as the model catalyst (Supplementary Fig. 33).

We then investigated CO₂ reduction to formaldehyde, nitrate reduction to hydroxylamine, and the subsequent hydrogenation steps following the coupling of hydroxylamine with formaldehyde (Fig. 6a–c and Supplementary Figs. 34–36). At zero applied potential (U = 0 V), the rate-limiting steps correspond to the hydrogenation of adsorbed *CHO during CO₂ reduction and *NO during NO₃⁻ reduction. Among the subsequent coupling and hydrogenation steps, the hydrogenation of *CH₂=NOH to *CH₃NOH is the most energetically demanding, although still thermodynamically favorable. Applying a potential of −0.95 V makes all hydrogenation steps increasingly exergonic, while a further decrease to −1.08 V provides an even larger thermodynamic driving force, thereby facilitating faster conversion of surface intermediates (Fig. 6a–c). Although −1.08 V shows the most negative free energies for proton-electron transfer steps within the computational hydrogen electrode framework, continuously applying such a deep potential is unlikely to maximize

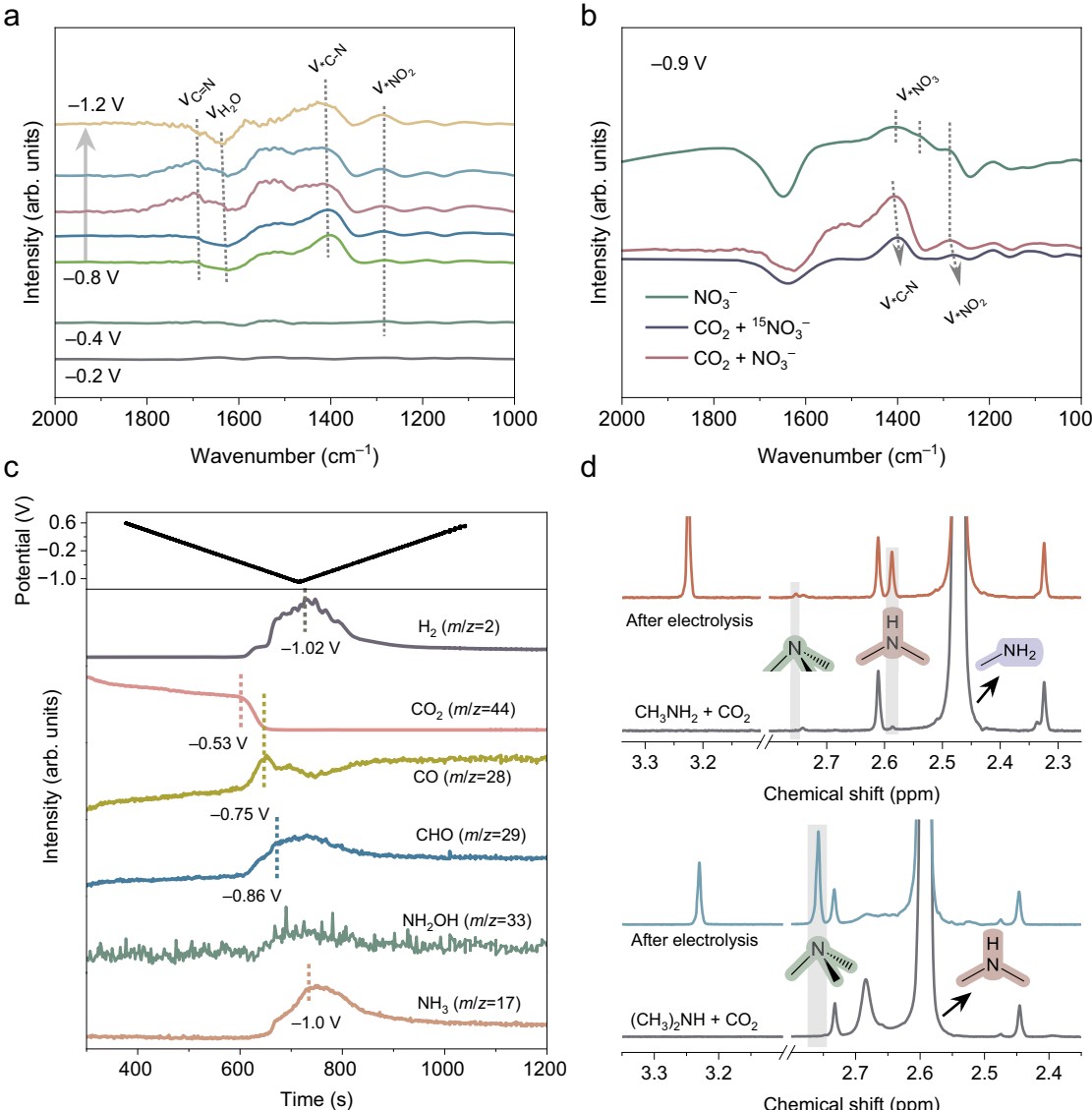

**Fig. 5 | Mechanism investigation. a** In-situ ATR-SEIRAS under different potentials in $CO_2$-saturated 0.1 M $KHCO_3$ and 0.5 M $KNO_3$. **b** Comparison of IR spectra at −0.9 V using Ar-saturated $K^{15}NO_3$, and $CO_2$-saturated $KHCO_3$ and $K^{15}NO_3/KNO_3$. **c** In-situ ECMS analysis for C−N coupling process. Potentials in (**a**−**c**) are reported without iR correction. **d** Retrosynthetic pathways from methylamine to dimethylamine and trimethylamine. Source data are provided with this paper.

selectivity. More negative potentials promote the conversion of oxime intermediates to amine products but also accelerate competing pathways, including hydrogen evolution and ammonia formation (Supplementary Fig. 16). These potential-dependent energetics support a pulsed strategy in which a moderate potential promotes the formation of formaldehyde and hydroxylamine intermediates, followed by a deeper reductive pulse that accelerates both the generation of C−N coupling intermediates and their hydrogenation to amine products.

To investigate the formation of the second C−N bond, we evaluated the Gibbs free energy changes for the coupling of formaldehyde with four possible intermediates: formaldoxime, methylhydroxylamine, methylenimine, and methylamine (Fig. 6d and Supplementary Fig. 37). The coupling of formaldoxime with formaldehyde is highly unfavorable and produces very unstable intermediates. The formation steps using methylamine and imine have Gibbs free energy changes of 0.04 eV, while methylhydroxylamine is slightly favorable (−0.01 eV), suggesting it is the most critical intermediate for dimethylamine formation. No imine-derived products are observed experimentally, indicating that methylenimine is unlikely to

participate under the reaction conditions. Nevertheless, given the high concentration of methylamine observed experimentally and supported by retrosynthetic experiments, methylamine also contributes to dimethylamine formation.

Based on these results, we propose a possible reaction mechanism. $NO_3^-$ is reduced via a six-electron process to hydroxylamine, while $CO_2$ undergoes a four-electron reduction to formaldehyde (Fig. 1c). Hydroxylamine couples with formaldehyde and dehydrates to form formaldoxime, which is hydrogenated to methylhydroxylamine and then to methylamine. Methylhydroxylamine and methylamine subsequently couple with formaldehyde to form the second C−N bond, followed by further hydrogenation to dimethylamine and eventually trimethylamine. Although selectivity and current density are still limited, pulsed electrolysis allows controlled hydrogenation and directs the reaction toward multialkylated amines, providing guidance for designing more selective catalysts and reaction conditions.

Techno-economic analysis (TEA, Supplementary Note 1) indicates that pulsed electrolysis provides a cost advantage over static operation for methylamine production, primarily due to improved selectivity and

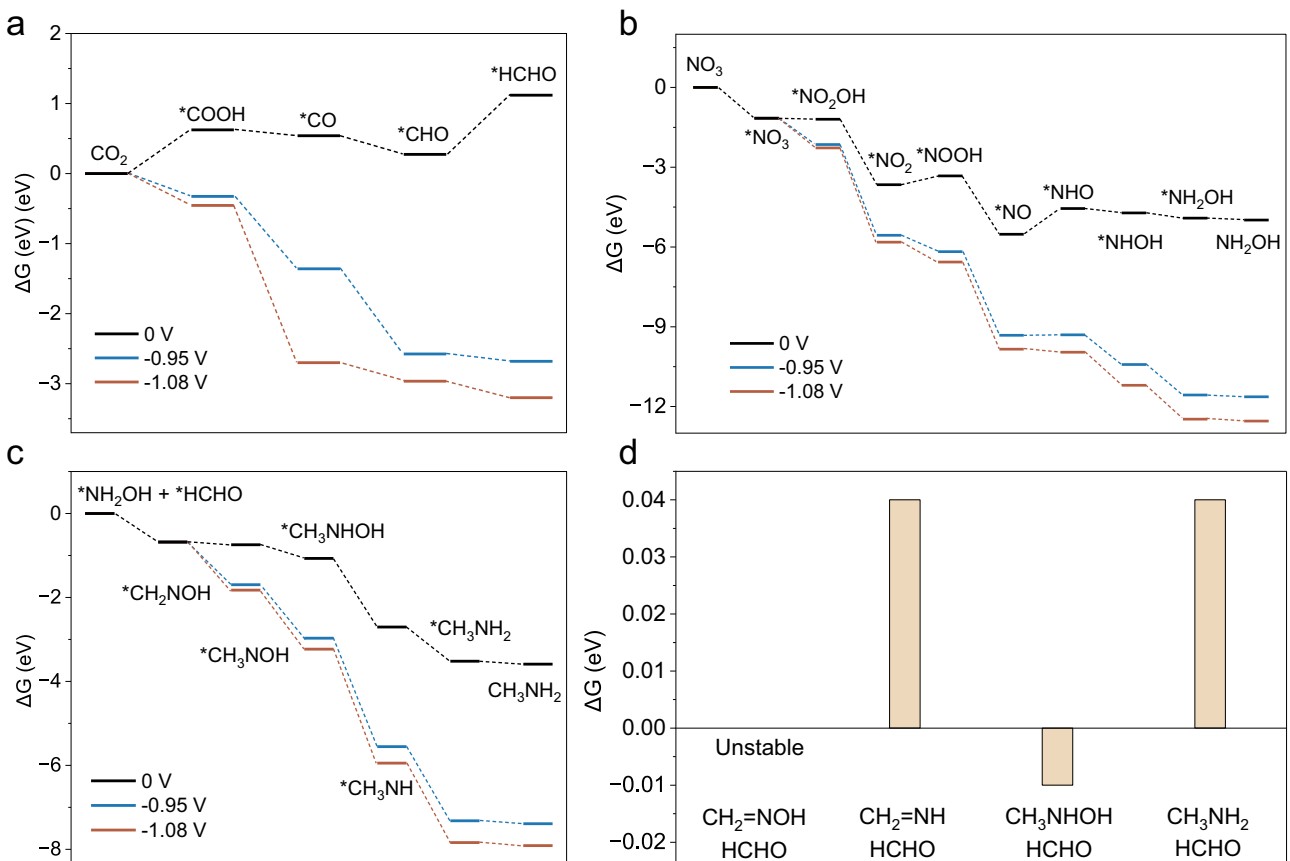

**Fig. 6 | Theoretical calculations of CO$_2$ and NO$_3^-$ reduction and subsequent C–N coupling. a** Free-energy changes (ΔG) of CO$_2$ reduction to HCHO* at different applied potentials (U = 0, −0.95, −1.08 V). **b** Free-energy profiles of NO$_3^-$ reduction to NH$_2$OH at different applied potentials. **c** Free-energy profiles for the coupling of NH$_2$OH* with HCHO* and the subsequent hydrogenation steps. **d** Calculated energy pathways for the formation of the second C–N bond, indicating CH$_3$NHOH as the key intermediate for the second C–N coupling. Source data are provided with this paper.

reaction rate, which increase product throughput and reduce energy consumption. Key cost drivers include FE, current density, electricity price, and stack cost (Supplementary Figs. 38, 39 and Tables 4, 5). These results highlight the importance of optimizing catalyst performance and operational parameters to enable scalable and economically viable electrochemical C–N coupling.

In summary, we report a pulsed electrolysis strategy to regulate hydrogenation during electrochemical C–N coupling using CO$_2$ and NO$_3^-$, enabling tunable access to oxime and amines. Systematic control of the pulsed potential reveals that less reductive pulse suppresses over-hydrogenation, favoring oxime formation, whereas more reductive pulse facilitates amine production. By alternating between a potential for C–N bond formation and more reductive potentials for hydrogenation, this approach enables high Faradaic efficiencies and current densities for methyl-, dimethyl-, and trimethyl- amines. Mechanistic investigations combining in situ ATR-SEIRAS, ECMS, retrosynthetic analysis, and DFT calculations reveal a stepwise hydrogenation pathway, in which deeper reduction potentials facilitate intermediate protonation and further conversion. Taken together, this work demonstrates that dynamic potential control offers a powerful tool to steer selectivity in electrosynthesis.

## Methods
### Chemicals
Potassium nitrate (KNO$_3$, 99%) was purchased from AppliChem GmbH. Potassium bicarbonate (KHCO$_3$, extra pure) and N, N-dimethylformamide (DMF, 99.99%) were purchased from Thermo

Fisher Scientific. Hydroxylamine hydrochloride (>97%), formoxime (10% in water), and Cobalt(II) phthalocyanine (CoPc, >93%) were purchased from TCI. $^{15}$N–labeled potassium nitrate ($^{15}$N-KNO$_3$, 98%), methanol (CH$_3$OH, 99.8%), dimethylamine (40 wt% in water), methylamine (40 wt% in water), formaldehyde solution (35–37%), multiwall carbon nanotubes ( ≥ 99.9 wt%), and $^{13}$CO$_2$ (98 atom %, <3 atom % $^{12}$C) were purchased from Sigma-Aldrich. N-methylhydroxylamine hydrochloride (97%) was purchased from BLD Chemicals. DMSO-d$_6$ (99.8%) and D$_2$O (99.9%) were purchased from Deutero GmbH. The anion exchange membrane (Fumasep FAA-3-PK-75) was purchased from Fuel Cell. Toray carbon paper (TGP-H-60) and graphite rod (99.9995%) were purchased from Thermo Fisher Scientific. The impedance of deionized (DI) water was 18.2 MΩ cm$^{-1}$. All chemical reagents were used without further purification.

### Material preparation and characterization
Commercial carbon nanotubes (CNTs) were calcined in air at 500 °C for 1 h with an airflow of approximately 50 mL min$^{-1}$. After cooling to room temperature, the material was immersed in an aqueous HCl solution (5 wt%) and sonicated for 30 min, followed by continuous stirring for 24 h. The resulting suspension was centrifuged, and the solid product was thoroughly washed with deionized water until the pH of the supernatant reached ~7. Subsequently, 30 mg of the purified CNTs was dispersed in 30 mL of N,N-dimethylformamide (DMF) via ultrasonication for 30 min. In parallel, 4.5 mg of CoPc was dissolved in 15 mL of DMF and ultrasonicated for 30 min. The two solutions were then combined, sonicated again for 60 min to ensure uniform mixing, and stirred at room temperature for 20 h. The

resulting suspension was centrifuged, and the collected precipitate was washed with DMF until the supernatant became clear, followed by washing with ethanol. Finally, the product was dried overnight in a vacuum oven.

X-ray diffraction (XRD) measurements were performed on Siemens Kristalloflex Diffraktometer D5000. TEM imaging was performed using a JEOL 1400 microscope operated at an acceleration voltage of 120 kV. Samples were prepared by drop-casting an ethanol dispersion of the catalyst onto a carbon-coated copper grid. XAS measurements were conducted at beamline P65 (PETRA III, DESY). UV–Vis absorption spectra were collected using an Agilent Cary 60 UV–Vis spectrophotometer. The inductively coupled plasma optical emission spectroscopy (ICP-OES) analysis was performed using the PerkinElmer Optima 8300, equipped with a Meinhard nebulizer and a glass cyclonic spray chamber. Electrospray ionization (ESI) in positive mode was measured with an Orbitrap XL instrument from Thermo Fisher Scientific. In-situ ATR-SEIRAS data were obtained with a Bruker Vertex 80 FTIR spectrometer.

### Electrochemistry and product quantification
To prepare the catalyst ink, 1 mg of CoPc/CNT was dispersed in 1 mL of ethanol containing 5 μL of 5 wt% Nafion solution, followed by ultrasonication for 1 h to ensure uniform dispersion. The carbon paper substrate (TGP-H-60, Fuel Cell Store) was pretreated by rinsing with water and drying under Ar to remove surface impurities. Subsequently, the catalyst ink was drop-cast onto a carbon paper substrate over a defined area of $0.5 \times 0.5$ cm$^2$ to achieve a catalyst loading of 0.3 mg cm$^{-2}$. After each drop, the electrode was completely dried under an infrared lamp.

Electrochemical measurements were performed using a potentiostat (SP50e, EC-Lab) in our home-made H-cell at ambient temperature (25 °C). The cathode consisted of carbon paper loaded with catalyst and the anode was a graphite rod. The cathodic and anodic compartments, each containing 5 mL of electrolyte, were separated by an anion exchange membrane (Fumsep FAA-3-PK-75, 70–80 μm thick), which was cut into a size of $2.5 \times 2.5$ cm$^2$. The membrane was pretreated in 0.1 M KOH for 12 h, thoroughly rinsed with deionized water, and subsequently equilibrated in 0.1 M KHCO$_3$ for 48 h prior to use. A working Ag/AgCl (sat. KCl) electrode was used and refilled with fresh saturated KCl solution before each test to ensure stability. Its potential was verified against a standard Ag/AgCl electrode (reserved only for calibration). Electrolytes were prepared by mixing deionized water with the requisite salts and used immediately after preparation. CO$_2$ was continuously supplied to the cathodic compartment at a flow rate of 20 mL min$^{-1}$ by an ALICAT digital mass flowmeter, and the electrolyte was continuously stirred at ca. 900 rpm. The ohmic resistance ($R_s$) was determined prior to each test using the Zero Input Resistance (ZIR) function. Based on this value, an 85% iR compensation was internally applied by the potentiostat for subsequent measurements (e.g., cyclic voltammetry and chronoamperometry). This procedure minimizes potential deviations caused by electrolyte resistance and ensures that the recorded potentials accurately reflect the interface conditions. The recorded current was normalized by the geometric surface area of the working electrode to obtain the current density (mA cm$^{-2}$). All potentials were converted to the reversible hydrogen electrode (RHE) scale using the following equation:

$$E\,(vs.\,RHE) = E\,(vs.Ag/AgCl) + 0.059*pH + 0.197V \tag{1}$$

The gas products were quantified by a gas chromatograph (8860 GC system, Agilent) equipped with a thermal conductivity detector and a flame ionization detector. The liquid products were quantified using $^1$H nuclear magnetic resonance (NMR) spectroscopy (500 MHz). Liquid products were quantified using the internal standard (IS)

method. In this approach, the product concentration is determined from the ratio of the integrated peak areas of the product and an internal standard with a known concentration. As many amine and oxime products are unstable and prone to volatilization, and isotopically labeled standards are unavailable for several species, the IS method was applied to ensure consistent quantification across all products. For $^1$H-NMR or $^{13}$C NMR analysis, 450 μl of electrolyte was mixed with 50 μl of 10 mM Maleic acid (99%, thermo scientific) in D$_2$O as the internal reference. The liquid products were quantified after electrocatalysis with water suppression.

For NH$_3$ quantification, 300 μL of electrolyte was mixed with 200 μL of 2 M H$_2$SO$_4$ and 100 μL of 10 mM maleic acid (99%, Thermo Scientific) in DMSO-d$^6$ (99.8%, Deutero). The NH$_3$ products were quantified after electrolysis using $^1$H NMR without water suppression.

For NH$_2$OH quantification, 500 μL of electrolyte was mixed with 2 μL of cyclohexanone and stirred for 5 min. Then, 450 μL of the resulting solution was combined with 50 μl of 10 mM Maleic acid in D$_2$O as an internal reference, and the products were analyzed by $^1$H NMR with water suppression.

The Faradaic efficiency (FE) and partial current density ($j$) of the liquid product were then calculated using the following equation[47]:

$$C_{analyte} = \frac{P_{analyte}*V_{internal}*H_{internal}*C_{internal}}{P_{internal}*V_{analyte}*H_{analyte}} \tag{2}$$

$$FE = \frac{C_{analyte}*V_{electrolyte}*z*F}{Q}*100\% \tag{3}$$

$$j = Q/(t*A)*FE \tag{4}$$

In the formulas, $C_{analyte}$ is the concentration of the analyte in the sample; $C_{internal}$ is the concentration of the internal standard; $P_{analyte}$ and $P_{internal}$ represent the NMR peak areas (integration) of the analyte and the internal standard, respectively; $V_{internal}$ is the volume of the internal standard added to the NMR sample, while $V_{analyte}$ is the volume of the analyte solution taken for the NMR measurement; $H_{internal}$ and $H_{analyte}$ denote the number of protons in the internal standard and the analyte contributing to the measured NMR peaks (e.g., CH$_3$ = 3, CH$_2$D = 2); $V_{electrolyte}$ refers to the total volume of the catholyte solution used in the electrolysis. In addition, $z$ is the number of electrons transferred per molecule of the analyte in the electrochemical reaction; $F$ is the Faraday constant (96485 C mol$^{-1}$); $Q$ is the total charge passed during the electrolysis; $t$ is the electrolysis time; $A$ is the geometric electrode area for catalysis.

The in-situ attenuated total reflection surface-enhanced infrared absorption spectroscopy (ATR-SEIRAS) measurements were carried out using an electrochemical cell. A gold (Au) film approximately 60 nm thick was sputtered onto the reflection plane of a Ge prism. Subsequently, catalyst ink was dropped onto the aforementioned film. A Pt foil served as the anode, and an Ag/AgCl electrode was used as the reference electrode. In-situ ECMS testing was carried out in a home-made electrochemical cell using CO$_2$-saturated 0.1 M KHCO$_3$ and 0.5 M KNO$_3$ electrolytes. A Pt foil served as the anode, and an Ag/AgCl electrode was used as the reference electrode. Cyclic voltammetry was performed at a scan rate of 5 mV s$^{-1}$ while simultaneously recording the mass spectrometry signals.

Density functional theory (DFT) calculations were carried out on a Vienna Ab initio Simulation Package (VASP 5.4.1)[48]. The exchange-correlation potential was described by the generalized gradient approximation (GGA) within the framework of Perdew-Burke-Ernzerhof (PBE) functional[49]. DFT-D3 method was employed to calculate the van der Waals (vdW) interaction[50]. The parameters of dipole correction were applied for the calculation of slab models. Electronic energies were computed with the tolerance of $1 \times 10^{-5}$ eV and total

force of 0.01 eV/Å. A kinetic cutoff energy of 450 eV was adopted. Carbon nanotube (CNT) with diameter of 10 nm was optimized and 252 C atoms fragment was chosen to simulate CNT. The Co-CNT slab was modeled by put the CoPc on the top of CNT fragment, and keeping the periodicity along y axis while inserting a vacuum space of 15 Å in x and z directions to avoid interactions between periodic images. To simulate the CNT effect, all the C atoms of CNT fragment were fixed during structure optimization. A Gamma-centered k-mesh of $1 \times 1 \times 1$ k-points were used in the structural relaxation for all models. The solvation effect was considered based on the implicit solvation model as implemented in VASPsol during all simulations[51,52]. However, these idealized models have limitations in capturing the full complexity of the real electrochemical interface.

The computational hydrogen electrode (CHE)[53] model was used to calculate the Gibbs free energy change ($\Delta G$) of $CO_2$ and $NO_3^-$ reduction reaction steps:

$$G = E_{DFT} + E_{ZPE} - TS \tag{5}$$

$$E_{ZPE} = \sum_i \frac{1}{2} h v_i \tag{6}$$

$$\Theta_i = h v_i / k \tag{7}$$

$$S = \sum_i R[\ln\left(1 - e^{-\frac{\Theta_i}{T}}\right)^{-1} + \frac{\Theta_i}{T}(e^{\frac{\Theta_i}{T}} - 1)^{-1}] \tag{8}$$

$$G[H^+ + e^-] = \frac{1}{2} G[H_2] + eU - 0.0592 * pH \tag{9}$$

where $E_{DFT}$ is the electronic energy calculated for specified geometrical structures, $E_{ZPE}$ is the zero-point energy, $S$ is the entropy, $h$ is the Planck constant, $v$ is the computed vibrational frequencies, $\Theta$ is the characteristic temperature of vibration, $k$ is the Boltzmann constant, and $R$ is the molar gas constant. * is the substrate, $U$ is the applied overpotential, and $e$ is the elementary charge. For adsorbates, all 3N degrees of freedom were treated as frustrated harmonic vibrations with negligible contributions from the catalysts' surfaces[54]. The optimized structures of all intermediates involved in this study were obtained from DFT calculations. The corresponding structural files are provided in Supplementary Data 1.

## Data availability
All data will be available upon request to the corresponding author via mail for non-commercial purposes. Data will be saved for 10 years, and requests will be responded to within 10 working days. Data corresponding to the main text and supplementary information are deposited alongside the manuscript as source data files. Source data are provided with this paper.

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

## Acknowledgements

We acknowledge the University of Bonn and the Deutsche Forschungsgemeinschaft grant KO 7060/4–1. Furthermore, we acknowledge DESY (Hamburg, Germany), a member of the Helmholtz Association HGF, for the provision of experimental facilities. Parts of this research were carried out at PETRA III. The Supercomputing Center of Max Planck Computing & Data Facility (MPCDF) is acknowledged for computational support.

## Author contributions

S.Y. conceived the project and conducted the experiments. Y.W. and M.M. provided technical support and scientific insight. S.C. contributed to the XAS experiment and analysis. J.H. assisted with TEA studies. C.G. contributed to SEM experiments and analysis. X.L. supervised the theoretical calculations and revised the manuscript. N.K. supervised the research.

## Funding

## Competing interests

The authors declare no competing interests
