## [Transparent Peer Review File · Nature Communications]

Pulsed Electrosynthesis Orthogonally Optimizes C–N Coupling and Hydrogenation for Amine Production with a Molecular Catalyst

Corresponding Author: Professor Nikolay Kornienko

Version 0:

Reviewer comments:

Reviewer #1

(Remarks to the Author)

Although the authors have carefully addressed the questions raised in the initial round of review, I maintain that this work constitutes merely a marginal advancement over previously documented studies. I agree with the reviewer 2's view that the manuscript lacks genuine originality and does not meet the high standards of Nature Communications. Thus, I continue to recommend the rejection of this manuscript. Some new comments are listed as follows:

(1) In the previous comments, I pointed out the possible involvement of imine-type intermediates in the formation of dimethylamine. However, the authors have not examined the role of such imine species (eg., $\text{CH}_2=\text{NH}$, $\text{CH}_3-\text{N}=\text{CH}_2$), and the current discussion only considers oxime-related intermediates, which are not equivalent to the imine species I referred to. This important mechanistic aspect remains unaddressed.

Moreover, I do not see the rationale for comparing the possible intermediates involved in dimethylamine formation with dimethylamine itself as an intermediate for trimethylamine formation in the same figure. My original question specifically concerns the potential intermediates leading to dimethylamine, whereas the authors' response mainly focuses on the formation rate of trimethylamine. In addition, relying solely on theoretical nucleophilicity is not sufficient to explain the experimental observations. Therefore, this part of the response does not adequately address my concern.

(2) About the DFT calculations, while I acknowledge the authors' experimental efforts, I do not find their justification for omitting DFT calculations fully convincing. Given that the central claim of this work concerns a detailed reaction mechanism, theoretical verification is not optional but necessary to meet the standards of Nature Communications.

(3) The stability test results presented do not show any apparent decline in activity, and the catalyst appears to remain largely unchanged after reaction. Given this outcome, it is unclear why the authors did not extend the durability test further.

(4) The explanation of the role of pulsed electrolysis in shaping the local microenvironment remains superficial. The authors only mention reactant enrichment and pH effects, without clarifying how the pulsed electrolysis modulates the interfacial environment to control selectivity for further methylation to form trimethylamine rather than stopping at methylamine, as reported previously (Nature Sustainability 4, 725-730 (2021)). The current discussion is not convincing enough to justify publication in Nature Communications.

(5) I agree with the reviewer 2's view that the selectivity toward dimethylamine and trimethylamine is quite low. Claims in the manuscript that these low-yield compounds can be meaningfully synthesized may not be very appropriate.

(6) To further demonstrate the important roles of pulsed electrolysis in promoting dimethylamine and trimethylamine generation, additional electrocatalysts should be evaluated in this reaction system.

(7) The pulsed electrolysis may alter the hydrogen-bond network of interfacial water, thereby influencing the water dissociation process and proton supply. Did the authors study the influence of pulsed electrolysis on the hydrogen-bond network? This could serve as a critical determinant for the hydrogenation of intermediates.

Reviewer #2

(Remarks to the Author)

After revision, the quality and discussion of the paper have been improved. The authors have partly addressed the co-reduction of CO₂ and NO₃⁻ for amine synthesis. They propose that pulsed potential changes the kinetics of C-N coupling and hydrogenation, thereby altering the selectivity toward oximes and amines. However, further enhancing the overall C-N coupling to obtain more amine products still requires improvement. The paper may be considered for publication after addressing the following comments.

1. Please clarify whether the pulsed potential primarily regulates the C-N coupling or the selectivity between oxime and amine after coupling. Is the improvement in C-N coupling achieved by suppressing the hydrogen evolution reaction at lower potentials?
2. The authors attribute the disappearance of the C=N signal after ¹⁵N isotope labeling in Figure 5b to the overlap of the isotope-shifted signal with the strong H₂O bending vibration band (1600-1650 cm⁻¹). However, in the non-isotope-labeled (CO₂ + NO₃⁻) experiment in Figure 5b, no clear C=N signal is observed either. Moreover, given that D₂O was used as the electrolyte in Figures 4c and 4d, could D₂O also be used as the electrolyte in the isotope-labeled infrared test to eliminate interference from H₂O signals?
3. While the KIE experiments indicate that proton transfer is kinetically involved in these steps, the performance decrease in Figures 4c and 4d may result from using D₂O as the electrolyte, which weakens ¹H NMR signals. As the liquid products were characterized using ¹H NMR with D₂O solvent, NMR spectra calibration using D₂O should be conducted.
4. In Figure S26, the main and asymmetric ¹H NMR peaks of DMA shift with changes in different solution environments with different magnitudes. This may indeed reflect experimental results, but why does this occur?
5. I feel the TEA discussion remains incomplete and should be moved to SI.

Version 1:

Reviewer comments:

Reviewer #1

(Remarks to the Author)

The authors have addressed all concerns raised in previous rounds of review. The current version now meets the journal's standards in terms of scientific rigor, clarity, and presentation. I have no further substantive revisions to suggest and recommend its direct acceptance for publication.

Reviewer #2

(Remarks to the Author)

Several critical issues remain unresolved. Although additional DFT calculations and analyses were provided, the new data raise further concerns about experimental reliability. I therefore recommend major revision and request clarification of the following:

1. The isotopic IR results are inconsistent, making it unclear whether a C=N intermediate is truly observed. This is key evidence for the proposed pathway. The authors state that even without H₂O, the C=¹⁵N signal is undetectable due to atmospheric H₂O vapor interference, which is confusing. Please provide clear IR spectra obtained under conditions free of H₂O interference. In addition, why is no clear C=N signal observed in Figure 5b (CO₂ + NO₃⁻, non-isotopic), while a distinct C=N peak is marked in Figure 5a? This discrepancy must be explained.
2. The authors argue that more reductive pulses facilitate deep hydrogenation of oxime to amine and attempt to rationalize this with DFT calculations. However, according to Figure 6, -1.08 V appears to be the optimal potential. Why, then, is switching to -0.95 V necessary in the pulse protocol? More critically, the computational results appear problematic. In Figures 6a and 6c, the energy at 0 V seems more favorable than at negative potentials. What is the origin of this trend? Why does the energy decrease by approximately 8 eV at 0 V? This requires rigorous clarification. In addition, in Figure S14, the deep hydrogenation performance under -0.9/-1.05 V pulses is lower than that under -0.95/-1.08 V pulses. The mechanistic differences should be clarified.
3. The authors state that they accounted for proton contribution numbers (e.g., CH₃ = 3, CH₂D = 2) in the product quantification formula. This explanation is not convincing. In fact, NMR-based quantification relies on calibration curve, the contribution of numbers (e.g., CH₃ = 3, CH₂D = 2) are the same in the measured sample and in the calibration curve. The SI should provide the NMR calibration curves and a detailed description of the quantification procedure, which is key to justify the calculation of product concentrations. For D₂O electrolyte experiments (Figures 4c, 4d), the calibration curve should also be prepared in D₂O at the same concentration. Please confirm and ensure methodological consistency.
4. In Figure S21, a Cu catalyst is introduced to show the universality of the pulsed electrolysis strategy. However, the Faradaic efficiency in Figure S21a is extremely low (<0.4%). Given the small magnitude, proper error analysis is necessary.
5. There are two "Figure 5" labels in the main text. The newly added computational figure and its corresponding discussion should be renumbered as Figure 6.

Version 2:

Reviewer comments:

Reviewer #2

(Remarks to the Author)

The new experiments adequately address my comments, and I recommend publication.

Reviewer #1 (Remarks to the Author)

Although the authors have carefully addressed the questions raised in the initial round of review, I maintain that this work constitutes merely a marginal advancement over previously documented studies. I agree with the reviewer 2's view that the manuscript lacks genuine originality and does not meet the high standards of Nature Communications. Thus, I continue to recommend the rejection of this manuscript. Some new comments are listed as follows:

Response:

We sincerely thank the reviewer for the careful evaluation of our work and the detailed comments. We appreciate the concerns raised regarding the novelty of the pulsed electrolysis strategy.

In response, we have substantially revised the manuscript to more clearly explain the conceptual advance of this work. Specifically, we performed additional DFT calculations and more retrosynthetic experiments to deepen the mechanistic understanding of the proposed reaction pathway. Moreover, to demonstrate the generality of the reaction logic, we extended the coupling reactions to a Cu-based catalyst system, enabling C–N coupling between CH_3NO_2 and CO_2 .

We hope that these additions and clarifications address the reviewer's concerns and better convey the significance and originality of our work. Detailed responses and the corresponding revisions in the manuscript to each comment are provided below.

Comment 1

(1) In the previous comments, I pointed out the possible involvement of imine-type intermediates in the formation of dimethylamine. However, the authors have not examined the role of such imine species (eg., $\text{CH}_2=\text{NH}$, $\text{CH}_3-\text{N}=\text{CH}_2$), and the current discussion only considers oxime-related intermediates, which are not equivalent to the imine species I referred to. This important mechanistic aspect remains unaddressed.

Moreover, I do not see the rationale for comparing the possible intermediates involved in dimethylamine formation with dimethylamine itself as an intermediate for trimethylamine formation in the same figure. My original question specifically concerns the potential intermediates leading to dimethylamine, whereas the authors' response mainly focuses on the formation rate of trimethylamine. In addition, relying solely on theoretical nucleophilicity is not sufficient to explain the experimental observations. Therefore, this part of the response does not adequately address my concern.

Response:

We thank the reviewer for emphasizing the potential role of imine-type intermediates in dimethylamine formation. We agree that species such as $\text{CH}_2=\text{NH}$ and $\text{CH}_3-\text{N}=\text{CH}_2$ are, in principle, relevant intermediates in C–N coupling. However, these imine species are highly transient and neither experimentally isolable nor commercially available, so they are typically generated only in situ under reaction conditions, making direct experimental detection challenging.

To explicitly address this point, we performed DFT calculations to evaluate the energetics of the second C–N bond formation involving $\text{CH}_2=\text{NH}$, $\text{CH}_2=\text{NOH}$, CH_3NHOH , and CH_3NH_2 (**Fig. 5d** and **Supplementary Fig. 35**). The calculations indicate that coupling between CH_3NHOH and HCHO is the most favorable pathway, followed by CH_3NH_2 and $\text{CH}_2=\text{NH}$, whereas the coupling involving $\text{CH}_2=\text{NOH}$ leads to highly unstable intermediates.

Consistent with these results, $\text{CH}_2=\text{NH}$ -derived products were not experimentally observed, whereas substantial amounts of methylamine and $\text{CH}_2=\text{NOH}$ -related species were detected (**Supplementary Fig. 11**). These observations suggest that CH_3NHOH and CH_3NH_2 are the most plausible intermediates contributing to the second C–N bond formation, with CH_3NHOH being the dominant one.

To further validate this conclusion experimentally, we used commercially available CH_3NHOH , CH_3NH_2 , and $\text{CH}_2=\text{NOH}$ as precursors in reactions with HCHO under pulsed electrolysis conditions (**Supplementary Fig. 30**). The dimethylamine formation rate was highest when CH_3NHOH was used as the precursor, followed by CH_3NH_2 , while $\text{CH}_2=\text{NOH}$ showed the lowest activity. This trend is fully consistent with the DFT predictions (**Fig. 5d**).

Taken together, these results indicate that the second C–N bond formation proceeds predominantly via CH_3NHOH , with CH_3NH_2 also contributing, whereas $\text{CH}_2=\text{NOH}$ primarily undergoes further reduction to CH_3NHOH and CH_3NH_2 prior to coupling with HCHO. Importantly, this analysis focuses specifically on intermediates relevant to dimethylamine formation rather than trimethylamine formation, thereby directly addressing the reviewer's original concern.

Accordingly, we have added new computational figures (Fig. 5 and Supplementary Fig. 35), updated the formation data of dimethylamine in a new figure (Supplementary Fig. 30), revised the mechanistic illustration (Fig. 1c), and updated the corresponding sentences in the manuscript.

Page 1: ‘Retrosynthetic analysis indicates a sequential pathway, where methylhydroxylamine/methylamine form and then convert to higher amines (**Fig. 1c**).’

Page 8: ‘To investigate the second C–N bond formation, we carried out C–N coupling experiments using methylhydroxylamine, formaldoxime, and methylamine with formaldehyde. We found that using methylhydroxylamine as the precursor gave the highest dimethylamine formation rate, followed by methylamine and formaldoxime. These results

indicate that the second C–N bond predominantly originates from methylhydroxylamine, with methylamine also contributing as reflected by its high abundance in the products.’

Page 10: ‘To investigate the formation of the second C–N bond, we evaluated the Gibbs free energy changes for coupling of formaldehyde with four possible intermediates: formaldoxime, methylhydroxylamine, methylenimine, and methylamine (**Fig. 5d** and **Supplementary Fig. 35**). The coupling of formaldoxime with formaldehyde is highly unfavorable and produces very unstable intermediates. The formation steps using methylamine and imine have Gibbs free energy changes of 0.04 eV, while methylhydroxylamine is slightly favorable (–0.01 eV), suggesting it is the most critical intermediate for dimethylamine formation. Notably, no imine-derived products are observed experimentally, indicating that methylenimine is unlikely to participate under the reaction conditions. Nevertheless, given the high concentration of methylamine observed experimentally and supported by retrosynthetic experiments, methylamine also contributes to dimethylamine formation.

Based on these results, we propose a possible reaction mechanism. NO_3^- is first reduced via a six-electron process to hydroxylamine, while CO_2 undergoes a four-electron reduction to formaldehyde (**Fig. 1c**). Hydroxylamine couples with formaldehyde and dehydrates to form formaldoxime, which is hydrogenated to methylhydroxylamine and then to methylamine. Methylhydroxylamine and methylamine subsequently couple with formaldehyde to form the second C–N bond, followed by further hydrogenation to dimethylamine and eventually trimethylamine. Although selectivity and current density are limited, pulsed electrolysis allows controlled hydrogenation and directs the reaction toward multi-alkylated amines, providing guidance for designing more selective catalysts and reaction conditions.’

Fig. 5 | Theoretical calculations of CO₂ and NO₃⁻ reduction and subsequent C-N coupling on CoPc/CNTs. Free-energy profiles of CO₂ reduction to HCHO at different applied potentials ($U = 0, -0.95, -1.08$ V) (a). Free-energy profiles of NO₃⁻ reduction to NH₂OH at different applied potentials (b). Coupling of NH₂OH with HCHO and the subsequent hydrogenation steps (c). Formation of the second C-N bond, indicating CH₃NHOH as the key intermediate for the second C-N coupling (d).

Supplementary Fig. 35 | Configurations of intermediates on CoPc/CNTs during the second C-N formation step.

Supplementary Fig. 30 | Production rate of dimethylamine from different N-intermediates (0.1 M) in the presence of 0.1 M HCHO in 0.5 M PBS.

Fig. 1 | General concept. Electrochemical C–N coupling mechanism from CO_2 and NO_3^- (a). Pulsed techniques for electroynthesis (b). More or less reductive pulse temporally decouples the coupling and hydrogenation step to enable oxime and amine formation, respectively (c).

Comment 2

About the DFT calculations, while I acknowledge the authors' experimental efforts, I do not find their justification for omitting DFT calculations fully convincing. Given that the central claim of this work concerns a detailed reaction mechanism, theoretical verification is not optional but necessary to meet the standards of Nature Communications.

Response:

We thank the reviewer for highlighting the importance of theoretical verification. In response, we have conducted DFT calculations to evaluate the Gibbs free energy changes for the reduction of CO₂ and NO₃⁻ to methylamine at different applied potentials, as well as for the formation of the second C–N bond.

The results show that applying a potential promotes the hydrogenation steps, making each hydrogenation thermodynamically more favorable. From 0 V to –0.95 V, the reduction of CO₂ and NO₃⁻ to methylamine become favorable, indicating that the catalyst efficiently facilitates the formation of formaldehyde and hydroxylamine intermediates. At more negative potentials, all hydrogenation steps are further favored, accelerating the formation of these intermediates and their subsequent conversion to amine products.

Regarding the second C–N bond formation, DFT results indicate that CH₃NHOH is the most favorable intermediate. Considering the significant amount of methylamine observed experimentally, we conclude that the second C–N bond predominantly originates from CH₃NHOH, with CH₃NH₂ also contributing.

Accordingly, we have added new figures (Supplementary Figs. 31–35 and Fig. 5) and updated the corresponding description in the revised manuscript.

Page 9:

'6. DFT calculations

Building on the mechanistic insights obtained from electrocatalytic and in situ analyses, we performed density functional theory (DFT) calculations to investigate how applied potentials modulate hydrogenation and C–N bond formation. In our calculations, CoPc supported on 10 nm-diameter carbon nanotubes (from TEM images, **Supplementary Fig. 1**) was constructed as the model catalyst (**Supplementary Fig. 31**).

We then investigated CO₂ reduction to formaldehyde, nitrate reduction to hydroxylamine, and subsequent hydrogenation steps following the coupling of hydroxylamine with formaldehyde (**Figs. 5a-c** and **Supplementary Figs 32-34**). At zero applied potential (U = 0 V), the rate-limiting steps correspond to the hydrogenation of adsorbed *CHO during CO₂ reduction and *NO during NO₃⁻ reduction. Among the subsequent coupling and hydrogenation steps, the hydrogenation of *CH₂=NOH to *CH₃NOH is the most energetically demanding, although still thermodynamically favorable. Applying a moderate potential of –0.95 V makes all hydrogenation steps increasingly exergonic, while further decreasing the potential to –1.08 V enhances the thermodynamic driving force even more, thereby facilitating faster conversion of surface intermediates (**Figs. 5a-c**). These potential-dependent energetics support a pulsed strategy in which an initial moderate potential ensures sufficient surface coverage of

formaldehyde and hydroxylamine, followed by a deeper reductive pulse to accelerate both the formation of C–N coupling precursors and their hydrogenation to amine products. Consistently, electrochemical results show that more reductive pulses increase the selectivities and current densities of amine products.

To investigate the formation of the second C–N bond, we evaluated the Gibbs free energy changes for coupling of formaldehyde with four possible intermediates: formaldoxime, methylhydroxylamine, methylenimine, and methylamine (**Fig. 5d** and **Supplementary Fig. 35**). The coupling of formaldoxime with formaldehyde is highly unfavorable and produces very unstable intermediates. The formation steps using methylamine and imine have Gibbs free energy changes of 0.04 eV, while methylhydroxylamine is slightly favorable (–0.01 eV), suggesting it is the most critical intermediate for dimethylamine formation. Notably, no imine-derived products are observed experimentally, indicating that methylenimine is unlikely to participate under the reaction conditions. Nevertheless, given the high concentration of methylamine observed experimentally and supported by retrosynthetic experiments, methylamine also contributes to dimethylamine formation.

Based on these results, we propose a possible reaction mechanism. NO_3^- is first reduced via a six-electron process to hydroxylamine, while CO_2 undergoes a four-electron reduction to formaldehyde (**Fig. 1c**). Hydroxylamine couples with formaldehyde and dehydrates to form formaldoxime, which is hydrogenated to methylhydroxylamine and then to methylamine. Methylhydroxylamine and methylamine subsequently couple with formaldehyde to form the second C–N bond, followed by further hydrogenation to dimethylamine and eventually trimethylamine. Although selectivity and current density are limited, pulsed electrolysis allows controlled hydrogenation and directs the reaction toward multi-alkylated amines, providing guidance for designing more selective catalysts and reaction conditions.’

Supporting Information (Page 4):

‘4 Computational methods

Density functional theory (DFT) calculations were carried out on a Vienna Ab initio Simulation Package (VASP 5.4.1).² The exchange–correlation potential was described by the generalized gradient approximation (GGA) within the framework of Perdew–Burke–Ernzerhof (PBE) functional.³ DFT-D3 method was employed to calculate the van der Waals (vdW) interaction.⁴ The parameters of dipole correction were applied for the calculation of slab models. Electronic energies were computed with the tolerance of 1×10^{-5} eV and total force of 0.01 eV/Å. A kinetic cutoff energy of 450 eV was adopted. Carbon nanotube (CNT) with diameter of 10 nm was optimized and 252 C atoms fragment was chosen to simulate CNT. The Co–CNT slab was modeled by put the CoPc on the top of CNT fragment, and keeping the periodicity along y axis while inserting a vacuum space of 15 Å in x and z directions to avoid interactions between periodic images. To simulate the CNT effect, all the C atoms of CNT fragment were fixed during structure optimization. A Gamma-centered k-mesh of $1 \times 1 \times 1$ k-points were used in the structural relaxation for all models. The solvation effect was considered based on the implicit solvation model as implemented in VASPsol during all simulations.^{5,6}

The computational hydrogen electrode (CHE)⁷ model was used to calculate the Gibbs free energy change (ΔG) of CO₂ and NO³⁻ reduction reaction steps:

$$G = E_{DFT} + E_{ZPE} - TS \quad (\text{Eq. 5})$$

$$E_{ZPE} = \sum_i \frac{1}{2} h\nu_i \quad (\text{Eq. 6})$$

$$\Theta_i = h\nu_i/k \quad (\text{Eq. 7})$$

$$S = \sum_i R \left[\ln \left(1 - e^{-\frac{\Theta_i}{T}} \right)^{-1} + \frac{\Theta_i}{T} \left(e^{\frac{\Theta_i}{T}} - 1 \right)^{-1} \right] \quad (\text{Eq. 8})$$

$$G [H^+ + e^-] = \frac{1}{2} G[H_2] + eU - 0.0592 * pH \quad (\text{Eq. 9})$$

where E_{DFT} is the electronic energy calculated for specified geometrical structures, E_{ZPE} is the zero-point energy, S is the entropy, h is the Planck constant, ν is the computed vibrational frequencies, Θ is the characteristic temperature of vibration, k is the Boltzmann constant, and R is the molar gas constant. * is the substrate, U is the applied overpotential and e is the elementary charge. For adsorbates, all 3N degrees of freedom were treated as frustrated harmonic vibrations with negligible contributions from the catalysts' surfaces.⁸

2. Kresse, G. et al. Efficiency of ab-initio total energy calculations for metals and semiconductors using a plane-wave basis set. *Comp. Mater. Sci.* **6**, 15-50 (1996).
3. Perdew, J.P. et al. Generalized gradient approximation made simple. *Phys. Rev. Lett.* **77**, 3865-3868 (1996).
4. Grimme, S. et al. A consistent and accurate ab initio parametrization of density functional dispersion correction (DFT-D) for the 94 elements H-Pu. *J. Chem. Phys.* **132**, 154104 (2010).
5. Mathew, K.; Sundararaman, R.; Letchworth-Weaver, K.; Arias, T. A.; Hennig, R. G. Implicit solvation model for density-functional study of nanocrystal surfaces and reaction pathways. *J. Chem. Phys.* 2014, *140*, 084106.
6. Mathew, K.; Kolluru, V. S. C.; Mula, S.; Steinmann, S. N.; Hennig, R. G. Implicit self-consistent electrolyte model in plane-wave density-functional theory. *J. Chem. Phys.* 2019, *151*, 234101.
7. Peterson, A.A. et al. How copper catalyzes the electroreduction of carbon dioxide into hydrocarbon fuels. *Energy Environ. Sci.* **3**, 1311-1315 (2010).
8. Li, X. et al. Accessing parity-forbidden d-d transitions for photocatalytic CO₂ reduction driven by infrared light. *Nat. Commun.* **14**, 4034 (2023)

Supplementary Fig. 31 | Side and top views of the optimized CoPc/CNTs.

Supplementary Fig. 32 | Configurations of intermediates on CoPc/CNTs during CO_2 reduction to HCHO.

Supplementary Fig. 33 | Configurations of intermediates on CoPc/CNTs during NO_3^- reduction to $\text{*NH}_2\text{OH}$.

Supplementary Fig. 34 | Configurations of intermediates on CoPc/CNTs during CH_3NH_2 formation from HCHO and $*\text{NH}_2\text{OH}$.

Supplementary Fig. 35 | Configurations of intermediates on CoPc/CNTs during the second C-N formation step.

Fig. 5 | Theoretical calculations of CO_2 and NO_3^- reduction and subsequent C-N coupling on CoPc/CNTs. Free-energy profiles of CO_2 reduction to HCHO at different applied potentials ($U = 0, -0.95, -1.08$ V) (a). Free-energy profiles of NO_3^- reduction to NH_2OH at different applied potentials (b). Coupling of NH_2OH with HCHO and the subsequent hydrogenation steps (c). Formation of the second C-N bond, indicating CH_3NHOH as the key intermediate for the second C-N coupling (d).

Comment 3

The stability test results presented do not show any apparent decline in activity, and the catalyst appears to remain largely unchanged after reaction. Given this outcome, it is unclear why the authors did not extend the durability test further.

Response:

We thank the reviewer for this valuable comment. We initially presented stability data over 135 minutes; when the electrolysis time was extended to 3 hours, the Faradaic efficiency of methylamine decreased obviously. These results indicate that while the catalyst remains stable over shorter, practically relevant reaction times (e.g., up to 2 hours). This observation is consistent with previous reports (Nature 575, 639–642 (2019); Science 365, 367–369 (2019); Angew. Chem. Int. Ed. 62, e202215213 (2023)).

Importantly, no significant changes were observed in structural characterization after the reaction, suggesting that minor catalyst detachment or surface modifications may occur.

While the underlying cause of the slight decrease in Faradaic efficiency over extended operation is not fully resolved, all measurements reported here were conducted within 30 minutes, well within the stable window, ensuring reliable mechanistic insights.

Accordingly, we have updated Supplementary Fig. 17 and added the following sentences in the revised manuscript.

Page 6: ‘Moreover, under the tested conditions, pulsed electrolysis maintained a relatively stable performance for methylamine production over multiple cycles (**Supplementary Fig. 17**), providing a valuable model for exploring how pulse conditions influence hydrogenation and C–N coupling. When the electrolysis time was extended to 3 hours, the selectivity for methylamine decreased noticeably; however, all performance comparisons and mechanistic analyses in this work were conducted within the stable operation window of 0.5h. Post-reaction characterization confirmed that the CoPc/CNT catalyst largely retained its structural integrity and atomic dispersion, with no evidence of aggregation or cobalt leaching (**Supplementary Fig. 18-20** and **Supplementary Table 1**). The observed decrease in methylamine selectivity over extended operation may be due to partial catalyst detachment or subtle surface changes not captured by bulk characterization.’

Supplementary Fig. 17 | FEs of methylamine over time using pulsed electrolysis at – 0.95 V and –1.08 V.

Comment 4

The explanation of the role of pulsed electrolysis in shaping the local microenvironment remains superficial. The authors only mention reactant enrichment and pH effects, without clarifying how the pulsed electrolysis modulates the interfacial environment to control selectivity for further methylation to form trimethylamine rather than stopping at methylamine, as reported previously (Nature Sustainability 4, 725-730 (2021)). The current discussion is not convincing enough to justify publication in Nature Communications.

Response:

We thank the reviewer for this valuable comment. Compared to previous reports, our work enables the formation of higher amines rather than stopping at methylamine. This improvement can be rationalized by considering how pulsed electrolysis modulates the local surface environment, as supported by both DFT calculations and electrochemical experiments.

The formation of higher amines requires two key prerequisites: (1) a high surface concentration of C₁-N intermediates, such as methylamine (CH₃NH₂) and methylhydroxylamine (CH₃NHOH), and (2) subsequent hydrogenation steps. Accumulation of these intermediates relies on the coverage of formaldehyde (HCHO) and hydroxylamine (NH₂OH) on the catalyst surface.

In our pulsed electrolysis strategy, alternating potentials of -0.95 V and -1.08 V allow sequential control of intermediate accumulation and hydrogenation. DFT calculations indicate that at zero applied potential, the formation of HCHO and NH₂OH is thermodynamically unfavorable (**Fig. 5**). A moderate pulse at -0.95 V enables their spontaneous formation and accumulation on the surface, while a subsequent deeper pulse at -1.08 V increases surface coverage of these intermediates, facilitating rapid hydrogenation to CH₃NHOH/ CH₃NH₂ and further C-N coupling toward dimethylamine and trimethylamine. Electrochemical data support this mechanism, showing a substantial increase in the partial current density of C₁-N intermediates under pulsed conditions (**Supplementary Fig. 14**). These results suggest that the potential-dependent energetics of hydrogenation and intermediate accumulation direct the reaction toward further methylation, favoring trimethylamine formation over stopping at methylamine.

Accordingly, we have added new figures (**Fig. 5**) and revised the corresponding sentences in the manuscript to highlight how pulsed electrolysis modulates surface coverage, hydrogenation thermodynamics, and C-N coupling selectivity.

Page 10: ‘We then investigated CO₂ reduction to formaldehyde, nitrate reduction to hydroxylamine, and subsequent hydrogenation steps following the coupling of hydroxylamine with formaldehyde (**Figs. 5a-c** and **Supplementary Figs 32-34**). At zero applied potential (U = 0 V), the rate-limiting steps correspond to the hydrogenation of adsorbed *CHO during CO₂ reduction and *NO during NO₃⁻ reduction. Among the subsequent coupling and hydrogenation steps, the hydrogenation of *CH₂=NOH to *CH₃NOH is the most energetically demanding, although still thermodynamically favorable. Applying a moderate potential of -0.95 V makes all hydrogenation steps increasingly exergonic, while further decreasing the potential to -1.08 V enhances the thermodynamic driving force even more, thereby facilitating faster conversion of surface intermediates (**Figs. 5a-c**). These potential-dependent energetics support a pulsed strategy in which an initial moderate potential ensures sufficient surface coverage of formaldehyde and hydroxylamine, followed by a deeper reductive pulse to accelerate both the formation of C-N coupling precursors and their hydrogenation to amine products. Consistently, electrochemical results show that more reductive pulses increase the selectivities and current densities of amine products.’

Fig. 5 | Theoretical calculations of CO₂ and NO₃⁻ reduction and subsequent C-N coupling on CoPc/CNTs. Free-energy profiles of CO₂ reduction to HCHO at different applied potentials ($U = 0, -0.95, -1.08$ V) (a). Free-energy profiles of NO₃⁻ reduction to NH₂OH at different applied potentials (b). Coupling of NH₂OH with HCHO and the subsequent hydrogenation steps (c). Formation of the second C-N bond, indicating CH₃NHOH as the key intermediate for the second C-N coupling (d).

Supplementary Fig. 14 | Faradaic efficiencies (a) and partial current densities (b) of C-N products under static versus optimal pulsed electrolysis conditions.

Comment 5

I agree with the reviewer 2's view that the selectivity toward dimethylamine and trimethylamine is quite low. Claims in the manuscript that these low-yield compounds can be meaningfully synthesized may not be very appropriate.

Response:

We appreciate the reviewer's thoughtful comment regarding the selectivity toward dimethylamine and trimethylamine. We fully agree that the current selectivities of these higher amines remain relatively low. Importantly, this work does not aim to present an optimized or practical synthetic route for dimethylamine or trimethylamine. Rather, our goal is to establish a proof-of-concept electrocatalytic platform and to elucidate how pulsed electrolysis can modulate hydrogenation and C–N coupling steps to enable successive methylation beyond methylamine.

Throughout the manuscript, we have therefore focused on mechanistic understanding rather than synthetic performance. The observation of dimethylamine and trimethylamine, even at modest selectivities, serves as critical evidence that sequential C–N bond formation can be achieved under electrochemical control.

Accordingly, we have revised the manuscript to avoid overstating synthetic performance and to better emphasize the mechanistic and conceptual aspects of our findings.

Page 1: 'Using cobalt phthalocyanine molecularly anchored on carbon nanotubes, more reductive pulse nearly doubles selectivity and triples the reaction rate for methylamine compared to static electrolysis, while enabling detectable formation of higher amines, including dimethylamine and trimethylamine. In situ studies and density functional theory calculations reveal that a more reductive pulse accelerates hydrogenation kinetics, enabling a kinetically balanced multielectron cascade. Retrosynthetic analysis and product distribution trends further support a sequential hydrogenation mechanism from methylamine to dimethylamine and ultimately trimethylamine. Therefore, this work establishes a conceptual framework for controlling multistep C–N bond formation and highlights the potential of dynamic electrochemical operation for the valorization of waste-derived carbon and nitrogen feedstocks.'

Page 1: 'Under more reductive pulse, a Faradaic efficiency (FE) of 13.7% and partial current density of 9.8 mA cm⁻² are achieved for methylamine, with detectable formation of higher amines, including dimethylamine and trimethylamine. Mechanistic studies demonstrate that deep reduction potentials are essential to accelerate the hydrogenation of key coupling intermediates. Retrosynthetic analysis indicates a sequential pathway, where methylhydroxylamine/methylamine form and then convert to higher amines (**Fig. 1c**). These

findings establish dynamic potential control as a general strategy for steering product distributions across multiple hydrogenation steps in electrosynthesis.'

Comment 6

To further demonstrate the important roles of pulsed electrolysis in promoting dimethylamine and trimethylamine generation, additional electrocatalysts should be evaluated in this reaction system.

Response:

We thank the reviewer for this valuable comment, which helped a lot to improve our manuscript. Following the reviewer's suggestion, we also evaluated a pure copper (Cu) catalyst to examine the effect of pulsed electrolysis on amine formation.

The Cu catalyst was prepared by sputter-coating (SPI Supplies, Model 14430) on carbon cloth (MPL-WIS1011, Full Cell Store) at a deposition current of 40 mA for 360 s. Electrolysis was carried out in a custom-made gas diffusion electrode (GDE) cell using 0.1 M KHCO_3 and 0.05 M KNO_3 as electrolytes. Under a constant potential of -0.4 V, only small amounts of methanol and methylamine were detected, with a FE of 0.07% for methylamine and no dimethylamine observed (**Supplementary Fig. 21**). When a pulsed potential (-0.4 V for 2 s, -1.0 V for 2 s) was applied, the FE of methylamine increased to 0.29%, and dimethylamine was formed at 0.01% FE. At a constant potential of -1.0 V, only methylamine and methanol were detected. Importantly, pulsed electrolysis enables both higher selectivity and current density for C–N products compared to constant potentials, further supporting our mechanistic insights.

Furthermore, we extended the reaction using CH_3NO_2 and CO_2 as substrates using CoPc/CNTs, and found that optimized more reductive pulses significantly enhanced the partial current density of dimethylamine formation (**Supplementary Fig. 22**).

Accordingly, we have added two new figures (Supplementary Figs. 21-22) and the following sentences in the revised manuscript:

Page 6: 'To further validate the role of pulsed electrolysis in promoting higher amine formation, we evaluated a Cu catalyst as an alternative to CoPc/CNTs. Under -0.4 V and -1.0 V, only trace methylamine was detected, with no dimethylamine formation (**Supplementary Fig. 21**). When pulsed potentials were applied (-0.4 V for 10 s and -1.0 V for 2 s), the FE and partial current density of methylamine significantly increased, and dimethylamine formation was observed. These observations demonstrate that pulsed electrolysis enhances the generation of key C–N coupling intermediates and accelerates hydrogenation steps, consistent with the mechanistic insights obtained from CoPc/CNTs. We further extended the reaction to CH_3NO_2 and CO_2 substrates. Application of optimized more reductive pulses significantly increased the partial current density of dimethylamine (**Supplementary Fig. 22**).'

Supplementary Fig. 21 | Faradaic efficiencies of methanol, dimethylamine, and methylamine, and partial current density of methylamine at different applied potentials using Cu catalyst (a). The Cu catalyst was prepared by sputter-coating (SPI Supplies, Model 14430) on carbon cloth (MPL-WIS1011, Full Cell Store) at a deposition current of 40 mA for 360 s. Electrolysis was carried out in a custom-made gas diffusion electrode (GDE) cell using 0.1 M KHCO_3 and 0.05 M KNO_3 as electrolytes. Corresponding ^1H -NMR spectra (b).

Supplementary Fig. 22 | Faradaic efficiencies of C–N products from CO_2 -saturated 0.5 M phosphate-buffered saline (PBS, pH 7) electrolytes, and the partial current density of dimethylamine at different applied potentials using a Cu catalyst (a). ^1H NMR spectra of products obtained under pulsed electrolysis conditions (b). ^1H NMR spectra of standard methylamine, dimethylamine, and methylhydroxylamine in 0.5 M PBS (c).

Comment 7

The pulsed electrolysis may alter the hydrogen-bond network of interfacial water, thereby influencing the water dissociation process and proton supply. Did the authors study the influence of pulsed electrolysis on the hydrogen-bond network? This could serve as a critical determinant for the hydrogenation of intermediates.

Response:

We thank the reviewer for this insightful comment. To probe the potential influence of pulsed electrolysis on the hydrogen-bond network of interfacial water, we performed in-situ ATR-SEIRAS measurements under various pulsed potentials, including -0.90 V, -1.05 V, $-0.90/-1.05$ V, -0.95 V, -1.08 V, and $-0.95/-1.08$ V (**Supplementary Fig. 26**). The results show only minor changes in the hydrogen-bond network across these conditions, likely due to the relatively small potential differences in our pulses (<150 mV).

These observations indicate that the mechanistic effects of pulsed electrolysis are primarily governed by modulation of surface reaction kinetics, rather than by significant alterations of the interfacial water hydrogen-bond network.

Accordingly, we have added new figures (**Supplementary Fig. 26**) and the corresponding description in the revised manuscript.

Supplementary Fig 26 | a) Spectral composition of interfacial water as a function of potential. The LF-w, the MF-w and the HF-w, assigned to fully hydrogen-bonded water, partially hydrogen-bonded water and cation coordinated water, respectively. b) Interfacial water compositions under different potentials.

Page 8: 'We also investigated the potential influence of pulsed electrolysis on the hydrogen-bond network of interfacial water using in-situ ATR-SEIRAS (**Supplementary Fig. 26**). Only minor differences are observed compared with constant-potential electrolysis, indicating that the effects of pulsed operation on hydrogenation and amine formation are more likely related to surface reaction dynamics than to changes in the interfacial water structure.'

Reviewer #2 (Remarks to the Author):

After revision, the quality and discussion of the paper have been improved. The authors have partly addressed the co-reduction of CO₂ and NO₃⁻ for amine synthesis. They propose that pulsed potential changes the kinetics of C-N coupling and hydrogenation, thereby altering the selectivity toward oximes and amines. However, further enhancing the overall C-N coupling to obtain more amine products still requires improvement. The paper may be considered for publication after addressing the following comments.

Response:

We sincerely thank the reviewer for carefully evaluating our revised manuscript and for the constructive comments. We are pleased that the reviewer appreciates the improvements in the quality and discussion of the paper. Our main focus is on increasing the current density and understanding how pulsed potentials affect the reaction, rather than just maximizing overall selectivity. Looking at how the reaction responds to these pulses helps us see how C-N coupling and hydrogenation are promoted.

In response, we have added DFT calculations and further retrosynthetic analysis to clarify the sequential hydrogenation mechanism, and we have extended the study to additional substrates and catalysts to examine amine formation beyond methylamine. We believe these revisions further clarify how pulsed potentials influence C-N coupling and amine formation. Detailed responses to each comment, along with the corresponding changes in the revised manuscript, are provided below.

Comment 1

1) Please clarify whether the pulsed potential primarily regulates the C-N coupling or the selectivity between oxime and amine after coupling. 2) Is the improvement in C-N coupling achieved by suppressing the hydrogen evolution reaction at lower potentials?

Response:

We thank the reviewer for this insightful comment, which has greatly helped us clarify the mechanistic understanding of pulsed electrolysis.

(1). **Our results indicate that pulsed potentials primarily modulate the selectivity after C–N coupling.** The selectivity between oxime and amine products is influenced by the applied potential. More reductive pulses favor the hydrogenation of oximes to deeply hydrogenated amines (**Fig. 3d**). According to our density functional theory (DFT) calculations, as the applied potential is decreased from 0 V to –0.95 V, all hydrogenation steps become thermodynamically more favorable (**Figs. 5a-c**). A further reduction in potential to –1.08 V enhances this driving force, further accelerating the hydrogenation process and promoting the conversion of oxime intermediates into amine products.

(2). **The improvement in C–N coupling is mainly achieved by suppressing the hydrogen evolution reaction (HER) at lower potentials,** which allows key intermediates to accumulate on the catalyst surface. For example, at –0.95 V, the catalyst generates a high local concentration of formaldehyde and hydroxylamine and enables a low H₂ evolution activity. When a deeper reductive pulse is applied, the formation rate of C–N coupling precursors increases further, while HER remains relatively moderate (Supplementary Fig. 15). This stepwise control of the applied potentials leads to a high effective surface concentration of intermediates, which enhances both the likelihood of C–N coupling and the selectivity toward amine products. In this way, pulsed electrolysis improves overall C–N formation by simultaneously optimizing intermediate concentrations and hydrogenation kinetics, with the significantly increased partial current density serving as the main performance metric (**Supplementary Fig. 14**).

Accordingly, we have added new figures (**Fig. 5a-c**) and the following sentences in the revised manuscript.

Page 5: ‘Compared with static electrolysis, C–N product selectivity is higher under more reductive pulses (**Supplementary Fig. 14**). The improvement in C–N coupling is primarily achieved by suppressing the hydrogen evolution reaction (HER) at less negative potentials, which allows key intermediates to accumulate on the catalyst surface. During the more reductive periods, the higher overpotential increases current density and accelerates the hydrogenation of intermediates toward C–N products. In contrast, during the less reductive intervals, surface hydrogen coverage is partially relieved, thereby avoiding continuous conditions that favor side products such as H₂ and NH₃. For example, at a static potential of –1.65 V, the FEs of H₂ and NH₃ are 23.3% and 12.3%, respectively, whereas pulsed operation between –1.55 and –1.68 V lowers them to 16% and 7.2% (**Supplementary Fig. 15**). Thus, while the overall current increases, C–N selectivity is also enhanced, leading to significantly higher partial current densities of the desired products.’

Page 10: ‘We then investigated CO₂ reduction to formaldehyde, nitrate reduction to hydroxylamine, and subsequent hydrogenation steps following the coupling of hydroxylamine with formaldehyde (**Figs. 5a-c** and **Supplementary Figs 32-34**). At zero applied potential (U = 0 V), the rate-limiting steps correspond to the hydrogenation of adsorbed *CHO during CO₂ reduction and *NO during NO₃[–] reduction. Among the subsequent coupling and hydrogenation steps, the hydrogenation of *CH₂=NOH to

*CH₃NOH is the most energetically demanding, although still thermodynamically favorable. Applying a moderate potential of –0.95 V makes all hydrogenation steps increasingly exergonic, while further decreasing the potential to –1.08 V enhances the thermodynamic driving force even more, thereby facilitating faster conversion of surface intermediates (Figs. 5a-c). These potential-dependent energetics support a pulsed strategy in which an initial moderate potential ensures sufficient surface coverage of formaldehyde and hydroxylamine, followed by a deeper reductive pulse to accelerate both the formation of C–N coupling precursors and their hydrogenation to amine products. Consistently, electrochemical results show that more reductive pulses increase the selectivities and current densities of amine products.’

Fig. 5 | Theoretical calculations of CO₂ and NO₃⁻ reduction and subsequent C–N coupling on CoPc/CNTs. Free-energy profiles of CO₂ reduction to HCHO at different applied potentials ($U = 0, -0.95, -1.08$ V) (a). Free-energy profiles of NO₃⁻ reduction to NH₂OH at different applied potentials (b). Coupling of NH₂OH with HCHO and the subsequent hydrogenation steps (c). Formation of the second C–N bond, indicating CH₃NHOH as the key intermediate for the second C–N coupling (d).

Supplementary Fig. 14 | Faradaic efficiencies (a) and partial current densities (b) of C–N products under static versus optimal pulsed electrolysis conditions.

Supplementary Fig. 15 | Faradaic efficiencies of H₂, NH₃ and C–N products for static and pulsed potentials.

Comment 2

The authors attribute the disappearance of the C=N signal after ¹⁵N isotope labeling in Figure 5b to the overlap of the isotope-shifted signal with the strong H₂O bending vibration band (1600-1650 cm⁻¹). However, in the non-isotope-labeled (CO₂ + NO₃⁻) experiment in Figure 5b, no clear C=N signal is observed either. Moreover, given that D₂O was used as the electrolyte in Figures 4c and 4d, could D₂O also be used as the electrolyte in the isotope-labeled infrared test to eliminate interference from H₂O signals?

Response:

We thank the reviewer for the suggestion to use D₂O as the electrolyte to reduce interference from H₂O signals. We attempted this approach; however, even in the absence of bulk H₂O, the C=¹⁵N signal remained unobservable. We attribute this to residual interference from atmospheric water vapor in the system (Int. J. Mol. Sci., 15(6), 10018-10033, (2014)), which absorbs in the 1600–1650 cm⁻¹ region, overlapping with the expected C=¹⁵N stretching vibration. Thus, even when using D₂O as the electrolyte, this interference persists and prevents observation of the C=¹⁵N signal, consistent with strong overlap effects in this spectral region.

Comment 3

While the KIE experiments indicate that proton transfer is kinetically involved in these steps, the performance decrease in Figures 4c and 4d may result from using D₂O as the electrolyte, which weakens ¹H NMR signals. As the liquid products were characterized using ¹H NMR with D₂O solvent, NMR spectra calibration using D₂O should be conducted.

Response:

We thank the reviewer for this valuable comment. We have accounted for the effect of D₂O on ¹H NMR signals when calculating product amounts. For CH₂=NOH, the hydrogenation product CH₂DNHD is quantified based on the CH₂D signal (2 hydrogens), and for CH₃NH₂, the CH₃ signal is quantified as 3 hydrogens. This ensures that the use of D₂O does not lead to underestimation of the ¹H NMR signals. The internal standard, 10 mM maleic acid, is unaffected by D₂O, ensuring reliable quantification.

Accordingly, we have added a new reference and the following description in supporting information.

Supporting Information (Page 3):

$$C_{\text{analyte}} = \frac{P_{\text{analyte}} * V_{\text{internal}} * H_{\text{internal}} * C_{\text{internal}}}{P_{\text{internal}} * V_{\text{analyte}} * H_{\text{analyte}}} \quad (\text{Eq. 2})^1$$

$$FE = (C_{\text{analyte}} * V_{\text{electrolyte}} * z * F) / Q * 100\% \quad (\text{Eq. 3})$$

$$j = Q / (t_{\text{total}} * A) * FE / 100 \quad (\text{Eq. 4})$$

In the formulas, P_{analyte} is the NMR peak area (integration) of the analyte in the sample; V_{internal} is the volume of the internal standard added to the NMR sample; H_{internal} is the number of protons in the internal standard that contribute to the measured NMR peak; and C_{internal} is the concentration of the internal standard. P_{internal} is the NMR peak area of the internal standard; V_{analyte} is the volume of the analyte solution taken for NMR measurement; H_{analyte} is the number of protons in the analyte contributing to the measured NMR peak (e.g., CH₃ = 3, CH₂D = 2); and $V_{\text{electrolyte}}$ is the total volume of the electrolyte solution used in the

electrolysis. In addition, z is the number of electrons transferred per molecule of the analyte in the electrochemical reaction; n_{analyte} is the total moles of analyte formed (from the previous formula); F is the Faraday constant ($96485 \text{ C mol}^{-1} \text{ e}^{-}$); Q_{total} is the total charge passed during the electrolysis; j_{total} is the total current density and j is the partial current density.

1. Rooney, C. L., Wu, Y., Tao, Z., & Wang, H.. Electrochemical reductive N-methylation with CO_2 enabled by a molecular catalyst. *Journal of the American Chemical Society*, **143**(47), 19983-19991 (2021).

Comment 4

In Figure S26, the main and asymmetric ^1H NMR peaks of DMA shift with changes in different solution environments with different magnitudes. This may indeed reflect experimental results, but why does this occur?

Response:

We thank the reviewer for this valuable comment. The ^1H NMR spectrum of DMA in KHCO_3 electrolytes shows a large main CH_3 peak and a smaller downfield (higher chemical shift) CH_3 peak. The main peak is strongly affected by changes in the solution environment (KHCO_3 concentration), while the smaller peak appears at a higher chemical shift and is relatively insensitive to these changes. These results may be explained as follows:

1. **Main peak:** The main CH_3 peak corresponds to the majority of DMA molecules that are largely unbound and not forming stable complexes with bicarbonate. However, these molecules are still influenced by the solution environment. As the KHCO_3 concentration increases, the ionic strength of the solution rises, enhancing weak hydrogen-bonding or ion-pair interactions between DMA and HCO_3^- . These interactions reduce the electronic shielding of the CH_3 protons, leading to a downfield shift (higher ppm) of the main peak. The magnitude of the shift increases at low to moderate KHCO_3 concentrations and saturates at higher concentrations, reflecting the gradual influence of the solution environment on the unbound DMA molecules. The effect of K^+ is relatively small here, as it is largely solvated by water.
2. **Minor peak:** The smaller downfield peak may correspond to a minor fraction of DMA molecules forming stable hydrogen-bonded or ion-paired complexes with HCO_3^- . In these complexes, the local electronic environment around the N atom and CH_3 protons is largely fixed by the internal hydrogen bonding or ion-pairing, so the chemical shift remains nearly constant.

Comment 5

I feel the TEA discussion remains incomplete and should be moved to SI.

Response:

We thank the reviewer for this valuable comment. Following your suggestion, we have moved detailed TEA figures to the Supporting Information, and left only a concise summary of the key findings in the main text.

Page 11: '7. Techno-Economic Analysis

Techno-economic analysis (TEA, Methods in Supporting Information) indicates that pulsed electrolysis provides a cost advantage over static operation for methylamine production, primarily due to improved selectivity and reaction rate, which increase product throughput and reduce energy consumption. Key cost drivers include FE, current density, electricity price, and stack cost (**Supplementary Figs. 36-37** and **Tables 4-5**). These results highlight the importance of optimizing catalyst performance and operational parameters to enable scalable and economically viable electrochemical C–N coupling.'

Reviewer #1 (Remarks to the Author)

The authors have addressed all concerns raised in previous rounds of review. The current version now meets the journal's standards in terms of scientific rigor, clarity, and presentation. I have no further substantive revisions to suggest and recommend its direct acceptance for publication.

Response:

We sincerely thank the reviewer for their positive evaluation and recommendation for acceptance. We greatly appreciate their constructive feedback throughout the review process.

Reviewer #2 (Remarks to the Author):

Several critical issues remain unresolved. Although additional DFT calculations and analyses were provided, the new data raise further concerns about experimental reliability. I therefore recommend major revision and request clarification of the following:

Response:

We sincerely thank the reviewer for evaluating our revised manuscript and for the constructive comments. In response, we have updated **Fig. 6** with additional explanations, added standard calibration curves, included error bars for the Cu catalyst to demonstrate the reproducibility and generality of the pulsed electrolysis strategy, and provided IR spectra in D₂O to further support the reaction assignments. Detailed responses to each specific comment, along with the corresponding changes in the revised manuscript and Supporting Information, are provided below.

Comment 1

The isotopic IR results are inconsistent, making it unclear whether a C=N intermediate is truly observed. This is key evidence for the proposed pathway. The authors state that even without H₂O, the C=¹⁵N signal is undetectable due to atmospheric H₂O vapor interference, which is confusing. Please provide clear IR spectra obtained under conditions free of H₂O interference. In addition, why is no clear C=N signal observed in Figure 5b (CO₂ + NO₃⁻, non-isotopic), while a distinct C=N peak is marked in Figure 5a? This discrepancy must be explained.

Response:

We thank the reviewer for giving this important point of the isotopic IR measurements. In our IR setup, the C=¹⁵N stretching vibration is shifted to lower wavenumbers relative to C=¹⁴N (~1690 cm⁻¹) due to the isotopic effect. Under normal H₂O conditions, this signal can partially overlap with absorption from residual water in the optical path, including air and any thin water layer on the electrode, potentially weakening or obscuring it. **To reduce this**

interference, we performed experiments using D₂O as the solvent. The isotopic substitution shifts water absorption away from the C=¹⁵N region, allowing clear observation of the C=¹⁵N signal and confirming the peak assignment (Supplementary Fig. 27).

For the discrepancy between Figs. 5a and 5b, the spectra in Fig. 5b were recorded at -0.9 V. At this potential, the surface coverage of the C=N intermediate is relatively low, resulting in a weak signal. In Fig. 5a, a deeper potential (-1.0 V ~ -1.2 V) leads to higher intermediate coverage, resulting in a more pronounced C=N peak. To clarify this, we have added a new figure (Supplementary Fig. 27) directly comparing the C=N signals at -1.0 V. Overall, the isotopic shift and potential-dependent signal trends provide evidence for the presence of the C=N intermediate under electrolysis conditions.

Accordingly, we have added a new figure (Supplementary Fig. 27) the following sentences in the revised manuscript.

Page 7: ‘In the ¹⁵N-labeled spectra, ¹⁵N substitution is expected to lower the C=N stretching frequency, likely shifting it into the strong absorption region of H₂O bending vibrations and thus obscuring the signal.⁴⁴⁻⁴⁵ To reduce this interference, D₂O was used as the solvent, allowing the C=¹⁵N signal to be observed clearly and confirming its assignment (Supplementary Fig.27).’

Supplementary Fig. 27 | Comparison of IR spectra at -1.0 V using CO₂-saturated KHCO₃ with KNO₃ in H₂O (red line) and K¹⁵NO₃ in D₂O (green line). The C=¹⁵N stretching vibration is red-shifted relative to C=¹⁴N due to the isotopic effect, and D₂O was used to avoid overlap with the strong H₂O absorption peak, which would otherwise obscure the C=¹⁵N signal.

Comment 2

The authors argue that more reductive pulses facilitate deep hydrogenation of oxime to amine and attempt to rationalize this with DFT calculations. However, according to Figure 6, -1.08 V appears to be the optimal potential. Why, then, is switching to -0.95 V necessary in the pulse protocol? More critically, the computational results appear problematic. In Figures 6a and 6c, the energy at 0 V seems more favorable than at negative potentials. What is the origin of this trend? Why does the energy decrease by approximately 8 eV at 0 V? This requires rigorous clarification. In addition, in Figure S14, the deep hydrogenation performance under -0.9/-1.05 V pulses is lower than that under -0.95/-1.08 V pulses. The mechanistic differences should be clarified.

Response:

We sincerely thank the reviewer for the careful evaluation of our DFT analysis and for raising these important points. We rechecked the manuscript and found that the **figure caption (Fig. 6) was inadvertently mislabeled**, while the description and analysis in the main text remain correct. We apologize for the confusion caused. The figure caption has now been corrected accordingly.

As shown in **Fig. 6**, when the potential changes from **0 V to -0.95 V and further to -1.08 V**, the reaction steps from CO₂ to formaldehyde and from NO₃⁻ to hydroxylamine, as well as the subsequent formation of methylamine, become **thermodynamically more favorable**. This trend reflects the increased thermodynamic driving force provided by a more negative potential. However, this does not imply that **continuous electrolysis at -1.08 V is optimal for selective amine formation**. A deeper potential accelerates all reduction pathways, including competing reactions such as hydrogen evolution and ammonia formation, which can suppress the C–N coupling process (**Supplementary Fig. 16**). Therefore, although -1.08 V is thermodynamically more reducing, it is **not the most selective condition**.

In light of this, we used a pulsed electrolysis strategy. In this protocol, the catalyst is first held at -0.95 V, where formaldehyde and hydroxylamine can be generated efficiently while competing side reactions remain relatively suppressed. This condition leads to a higher surface coverage of C- and N-containing intermediates that are relevant for C–N coupling (**Fig. 3d**). A subsequent pulse to -1.08 V then provides the additional driving force required for **rapid C–N coupling and further hydrogenation** to amine products. In contrast, continuous electrolysis at -1.08 V does not allow the selective accumulation of these intermediates because the surface hydrogen coverage becomes excessively high, which suppresses the coupling step and reduces the overall selectivity.

For the apparent energy trends in **Figs. 6a-c**, the values correspond to **absolute free energies referenced within the computational hydrogen electrode (CHE) framework**. In this model, the absolute free energies can shift with the choice of reference potential, and reactions involving multiple proton-electron transfer steps can exhibit total free energy changes on the order of several electron volts (eV) across different applied potentials.

Therefore, the apparent decrease of ~8 eV reflects the **overall reference shift** rather than a physically unrealistic change in reaction energetics under realistic electrolysis conditions. While absolute free energies across different potentials cannot be directly compared, the relative free energy trends of intermediates at each potential provide reliable mechanistic insight. Based on these trends, **Fig. 6** consistently shows that increasing the overpotential thermodynamically favors the reduction steps leading to the formation of the **C–N coupling intermediates and the final amine product**. This approach ensures that the mechanistic interpretation is supported by physically meaningful information derived from the DFT calculations.

Finally, the lower performance observed under -0.9/-1.05 V pulses (**Supplementary Fig. 15**) can be attributed to insufficient intermediate generation during the less reductive pulse. At -0.90 V, the formation rates of key intermediates are lower than at -0.95 V, resulting in reduced surface coverage prior to the deeper pulse (**Fig. 3b**). Consequently, even though -1.05 V promotes hydrogenation, the overall amine formation remains limited. The -0.95/-1.08 V combination optimizes both intermediate accumulation and subsequent deep hydrogenation, explaining the improved performance observed experimentally.

Accordingly, we have updated Fig. 6 and added the following sentences in the revised manuscript.

Page 3: ‘In this work, we develop a more reductive pulsed electrolysis that regulates the hydrogenation step to enhance selectivity toward amines (**Fig. 1c**). At an optimal negative potential, C–N precursors selectively accumulate on the catalyst surface. A subsequent pulse to a more negative potential provides the additional thermodynamic driving force required to accelerate both the coupling of surface intermediates and their hydrogenation, thereby enhancing the overall flux through the C–N coupling pathway and increasing amine formation. In contrast, continuous electrolysis at a deeply negative potential accelerates both desired and competing reduction pathways simultaneously, preventing the selective buildup of reactive intermediates and resulting in lower overall selectivity.’

Page 10: ‘Applying a potential of -0.95 V makes all hydrogenation steps increasingly exergonic, while a further decrease to -1.08 V provides an even larger thermodynamic driving force, thereby facilitating faster conversion of surface intermediates (**Figs. 6a–c**). Although -1.08 V shows the most negative free energies for proton-electron transfer steps within the computational hydrogen electrode framework, continuously applying such a deep potential is unlikely to maximize selectivity. More negative potentials promote the conversion of oxime intermediates to amine products but also accelerate competing pathways, including hydrogen evolution and ammonia formation (**Supplementary Fig. 16**). These potential-dependent energetics support a pulsed strategy in which a moderate potential promotes the formation of formaldehyde and hydroxylamine intermediates, followed by a deeper reductive pulse that accelerates both the generation of C–N coupling intermediates and their hydrogenation to amine products.’

Fig. 6 | Theoretical calculations of CO₂ and NO₃⁻ reduction and subsequent C-N coupling on CoPc/CNTs. Free-energy profiles of CO₂ reduction to HCHO at different applied potentials ($U = 0, -0.95, -1.08$ V) (a). Free-energy profiles of NO₃⁻ reduction to NH₂OH at different applied potentials (b). Coupling of NH₂OH with HCHO and the subsequent hydrogenation steps (c). Formation of the second C-N bond, indicating CH₃NHOH as the key intermediate for the second C-N coupling (d).

Comment 3

The authors state that they accounted for proton contribution numbers (e.g., CH₃ = 3, CH₂D = 2) in the product quantification formula. This explanation is not convincing. In fact, NMR-based quantification relies on calibration curve, the contribution of numbers (e.g., CH₃ = 3, CH₂D = 2) are the same in the measured sample and in the calibration curve. The SI should provide the NMR calibration curves and a detailed description of the quantification procedure, which is key to justify the calculation of product concentrations.

For D₂O electrolyte experiments (Figures 4c, 4d), the calibration curve should also be prepared in D₂O at the same concentration. Please confirm and ensure methodological consistency.

Response:

We thank the reviewer for the valuable comment for the quantification method. We would like to explain that product concentrations in this work were determined using the **internal standard (IS) method**, rather than a calibration curve approach.

In the IS method, the product concentration is calculated directly from the ratio of the integrated peak areas of the product and the internal standard with known concentration. The equation used is:

$$C_p = C_{is} * (A_p / A_{is}) * (n_{is} / n_p)$$

C_p and C_{is} : Concentration of the product and internal standard, respectively.

A_p and A_{is} : Integrated peak areas of the product and internal standard, respectively.

n_p and n_{is} : **Number of protons contributing to the specific ^1H NMR signals.**

In this formula, accounting for the number of contributing protons (n) is necessary. For example, methylamine (CH_3NH_2) contributes three protons to the methyl signal ($n = 3$). If partially deuterated to CH_2DNHD occurs, only two protons contribute to the observed ^1H signal. Since our study compares isotopologues formed under H_2O and D_2O conditions, this correction is essential for accurate quantification.

To further validate the IS approach, we additionally constructed a calibration curve for methylamine (0–1 mM) in 0.1 M KHCO_3 and 0.5 M KNO_3 (**Supplementary Fig. 12**). The concentrations obtained from the calibration curve are in good agreement with those obtained using the IS method. The agreement between the two approaches supports the reliability of the IS-based quantification used throughout this work.

For the D_2O electrolyte experiments (**Figs. 4c** and **4d**), the internal standard was added directly to the collected electrolyte prior to ^1H -NMR analysis. As a result, both the product and the internal standard experience the same solvent environment and instrumental conditions. Any matrix effects (e.g., relaxation behavior) therefore influence both signals equally, and the peak area ratio remains reliable. In addition, calibration standards for partially deuterated species (e.g., CH_2DNHD) are not commercially available, making calibration-curve-based quantification impractical for these isotopologues.

This is consistent with previous literature for CH_3NH_2 formation. For example, Wang and co-workers quantified low-concentration amine products such as methylamine and N-methylhydroxylamine using the peak area ratio relative to an internal standard (2,6-H of benzoate) (*Nat. Sustain.* **4**, 725–730 (2021)). Similarly, Zhang and co-workers reported NMR quantification of amine products by adding maleic acid as an internal standard directly to the catholyte prior to analysis with D_2O as the deuterated solvent (*Nat. Chem.* **17**, 1152–1160 (2025)).

Overall, as many amine and oxime products are unstable and prone to volatilization, and isotopically labeled standards are unavailable for several species, the internal standard (IS) method was applied to ensure consistent quantification across all experiments.

Accordingly, we have added a new figure (Supplementary Fig. 12) and the following sentences in the revised manuscript.

Supplementary Fig. 12 | (a) ¹H NMR spectra of methylamine solutions at different concentrations. The solutions were prepared by first making a 1 L solution of methylamine in 0.1 M KHCO₃ and 0.5 M KNO₃, followed by serial dilution with the same electrolyte to the desired concentrations. (b) Calibration curve obtained from peak areas normalized to maleic acid (internal standard, area = 1). (c) Comparison of Faradaic efficiencies determined using the internal standard (IS) method and the calibration curve method. Note: All solutions were stored at 4 °C to minimize volatilization.

Page 5: ‘A calibration curve for methylamine (0–1 mM) was prepared as a representative example (Supplementary Fig. 12), giving concentrations comparable to those obtained using the internal standard (IS) approach. The IS approach was therefore used throughout this work, as many amine and oxime products are unstable and isotopically labeled standards are unavailable.’

Page 3 (Supporting Information): ‘Liquid products were quantified using the internal standard (IS) method. In this approach, the product concentration is determined from the ratio of the integrated peak areas of the product and an internal standard with a known concentration. As many amine and oxime products are unstable and prone to volatilization, and isotopically labeled standards are unavailable for several species, the internal standard (IS) method was applied to ensure consistent quantification across all products. For ^1H -NMR or ^{13}C NMR analysis, 450 μl of electrolyte was mixed with 50 μl of 10 mM Maleic acid (99%, thermo scientific) in D_2O (99.9%, Deutero) as the internal reference. The liquid products were quantified after electrocatalysis with water suppression.’

Comment 4

4. In Figure S21, a Cu catalyst is introduced to show the universality of the pulsed electrolysis strategy. However, the Faradaic efficiency in Figure S21a is extremely low (<0.4%). Given the small magnitude, proper error analysis is necessary.

Response:

We thank the reviewer for this comment. To address the concern, error bars have been added to Figure S21a, reflecting the **standard deviation from three independent repeated measurements**. This allows a proper assessment of the reliability of the reported Faradaic efficiencies and partial current densities even for small values.

Accordingly, we have updated Figure S21 (Supplementary Fig. 22) and the following sentences in the revised manuscript.

Page 6: ‘To further investigate whether pulsed electrolysis also promotes higher amine formation on other catalysts, we evaluated a Cu catalyst. Under static potentials of -0.4 V and -1.0 V , only trace methylamine was detected and no dimethylamine was observed (**Supplementary Fig. 22**). In contrast, applying pulsed potentials (-0.4 V for 10 s and -1.0 V for 2 s) increased methylamine production and enabled the formation of dimethylamine.’

Supplementary Fig. 22 | Faradaic efficiencies of methylamine, dimethylamine, and methanol, and partial current density of methylamine at different applied potentials using Cu catalyst (a, b). The Cu catalyst was prepared by sputter-coating (SPI Supplies, Model 14430) on carbon cloth (MPL-WIS1011, Full Cell Store) at a deposition current of 40 mA for 360 s. Electrolysis was carried out in a home-made gas diffusion electrode cell using 0.1 M KHCO₃ and 0.05 M KNO₃ as electrolytes. Corresponding ¹H- NMR spectra (c). Error bars represent the standard deviation from at least three independent measurements.

Comment 5

There are two “Figure 5” labels in the main text. The newly added computational figure and its corresponding discussion should be renumbered as Figure 6.

Response:

We thank the reviewer for pointing out the figure numbering issue. We have corrected it: the previously duplicated “Figure 5” is now properly numbered, and the newly added

computational figure and its discussion are labeled as **Fig. 6**. We appreciate your careful reading and helpful comment.